# The Shu complex interacts with the replicative helicase to prevent mutations and aberrant recombination

Adeola A Fagunloye [1,7], Alessio De Magis [2,3,7], Jordan H Little [4], Isabela Contreras[1],
Tanis J Dorwart [1], Braulio Bonilla [5], Kushol Gupta [1], Nathan Clark [4,6], Theresa Zacheja[3],
Katrin Paeschke [2,3,7 ✉] & Kara A Bernstein [1,7 ✉]

## Abstract

**Homologous recombination (HR) is important for DNA damage tolerance during replication. The yeast Shu complex, a conserved homologous recombination factor, prevents replication-associated mutagenesis. Here we examine how yeast cells require the Shu complex for coping with MMS-induced lesions during DNA replication. We find that Csm2, a subunit of the Shu complex, binds to autonomous-replicating sequences (ARS) in yeast. Further evolutionary studies reveal that the yeast and human Shu complexes have co-evolved with specific replication-initiation factors. The connection between the Shu complex and replication is underlined by the finding that the Shu complex interacts with the ORC and MCM complexes. For example, the Shu complex interacts, independent of other HR proteins, with the replication initiation complexes through the N-terminus of Psy3. Lastly, we show interactions between the Shu complex and the replication initiation complexes are essential for resistance to DNA damage, to prevent mutations and aberrant recombination events. In our model, the Shu complex interacts with the replication machinery to enable error-free bypass of DNA damage.**

**Keywords** Shu Complex; DNA Replication; Homologous Recombination; DNA Damage Tolerance; RAD51
**Subject Category** DNA Replication, Recombination & Repair

## Introduction

The evolutionarily conserved Shu complex functions to promote high-fidelity homologous recombination (HR) during DNA replication (Mankouri et al, 2007; Krejci et al, 2012; Bonner and Zhao, 2016; Godin et al, 2016a; Martino and Bernstein, 2016; Ball et al, 2009). The budding yeast Shu complex is an obligate heterotetramer composed of three Rad51 paralogs, Csm2, Psy3, Shu1, and a SWIM-domain containing protein, Shu2 (Shor et al, 2005; Xu et al, 2013; Ball et al, 2009). During the early steps of HR, the central recombinase, Rad51, forms nucleoprotein filaments on ssDNA thus enabling a homology search and subsequent strand invasion of an undamaged repair template (Sullivan and Bernstein, 2018). The formation of Rad51 filaments is tightly regulated to ensure that HR occurs in an accurate and timely fashion (Bonilla et al, 2020; Crickard and Greene, 2019). The Rad51 paralogs are ancient gene duplications of Rad51 that have evolved to regulate Rad51 function (Liu et al, 2011; Xu et al, 2013; Sasanuma et al, 2013; Hays et al, 1995; Maloisel et al, 2023). In yeast, the Rad51 paralogs include Rad55-Rad57, which have broad roles in HR compared to the more specialized Shu complex-containing Rad51 paralogs, Csm2, Psy3, and Shu1 (Xu et al, 2013; Ma et al, 2018; Maloisel et al, 2023; van Mourik et al, 2016). The Shu complex together with Rad55-Rad57 and the Rad51 loader, Rad52, form an ensemble of proteins that facilitate Rad51 filament assembly (Gaines et al, 2015; Sasanuma et al, 2013; Roy et al, 2021).

During DNA replication, DNA damage can be bypassed using a template switching mechanism that is facilitated by the recombinase, Rad51 (González-Prieto et al, 2013; Vanoli et al, 2010; Mojumdar et al, 2022). The yeast Shu complex facilitates the formation of Rad51 filaments in this replicative context where its function is restricted (Godin et al, 2016b; Gaines et al, 2015; González-Prieto et al, 2013). This is unique to other HR factors that repair direct DSBs outside of DNA replication. How the Shu complex function is limited to facilitate bypass of replicative DNA damage is enigmatic. However, hints come from its DNA damage sensitivity, where the loss of any Shu complex members results in sensitivity to the alkylating agent, methyl methanesulfonate (MMS) (Godin et al, 2016a, 2013; Shor et al, 2005; Ball et al, 2009; Mankouri et al, 2007; Huang et al, 2003). Partially explaining this specificity for replicative repair, the Shu complex DNA binding subunits, the Rad51 paralogs Csm2-Psy3, preferentially bind to

[1]University of Pennsylvania, School of Medicine, Department of Biochemistry and Biophysics, Philadelphia, PA 19104, USA. [2]Department of Oncology, Hematology and Rheumatology, University Hospital Bonn, Bonn, Germany. [3]Department of Clinical Chemistry and Clinical Pharmacology, University Hospital Bonn, Bonn, Germany. [4]University of Utah, Department of Human Genetics, Salt Lake City, UT 84112, USA. [5]University of Pittsburgh, School of Medicine, Department of Pharmacology and Chemical Biology, Pittsburgh, PA 15213, USA. [6]University of Pittsburgh, Department of Biological Sciences, Pittsburgh, PA 15260, USA. [7]These authors contributed equally: Adeola A Fagunloye, Alessio De Magis, Katrin Paeschke, Kara A Bernstein. ✉E-mail: kpaeschk@uni-bonn.de; kara.bernstein@pennmedicine.upenn.edu

double-flap substrates and have increased affinity for a double-flap containing an abasic site, which forms during repair of alkylation damage (Rosenbaum et al, 2019). Loss of Shu complex function results in translesion synthesis-induced mutations and the mutation rate increases over 1000-fold when abasic sites accumulate (Gaines et al, 2015; Godin et al, 2016b; Bonilla et al, 2021). Despite the fundamental connection between the Shu complex and replication, how the Shu complex is recruited to replication forks is unknown.

DNA replication begins at defined replication origins in yeast, which are located within autonomous replicating sequences [ARS; (Liachko and Dunham, 2014; Lee et al, 2023)]. In G1, the origin recognition complex (ORC), consisting of ORC 1-6, binds the replication origins (Da-Silva and Duncker, 2007). During origin licensing, the ORC complex and Cdc6 together with Cdt1 facilitate recruitment of the minichromosome maintenance 2-7 (MCM) complex and its loading onto dsDNA at replication origins (Yuan et al, 2017; Evrin et al, 2014; Schmidt et al, 2022). Subsequently, Cdc6 and Cdt1 facilitate the binding of a second MCM complex and ORC is released (Yuan et al, 2017; Evrin et al, 2014). Upon S phase, cyclin-dependent kinases (CDKs) and Dbf4-dependent kinases (DDKs) phosphorylate multiple pre-replication complex components, which enable Cdc45, the MCM complex, and the GINS (consisting of Sld5, Psf1, Psf2, and Psf3) to form the replicative helicase (CMG) (Yabuuchi et al, 2006; Lei and Tye, 2001). The CMG is associated with DNA polymerase epsilon and alpha to initiate leading and lagging strand DNA synthesis (Yabuuchi et al, 2006; Bauerschmidt et al, 2007; Lei and Tye, 2001). In addition to the factors mentioned above and the simplified model described, there are many other accessory proteins and other modifications that facilitate that origin licensing and firing. Together, these factors limit replication stress and enable high fidelity and timely genome duplication.

To address how the Shu complex promotes error-free bypass of DNA damage during S phase, we assess the binding of the primary Shu complex DNA binding subunit, Csm2. We find that Csm2 is enriched at ARS sites and that it is coevolving with the replication initiation machinery. We find that multiple Shu complex members interact with the ORC complex by yeast-2-hybrid (Y2H) and co-immunoprecipitation (coIP), and these interactions occur in G1 in a DNA-independent manner. The Shu complex also maintained interaction with the MCM complex in G1 and remains associated during S phase. We mapped the interface between the Shu complex protein, Psy3, with the ORC and MCM complexes, and showed that Psy3 N-terminus is highly conserved and mediates these protein interactions. Mutations in the invariant Y10 residue of Psy3 results in MMS sensitivity, increased mutations, and altered recombination. Together, our work provides insights into the mechanism by which the Shu complex safeguards genomic integrity during DNA replication.

# Results

## The Shu complex DNA binding subunit, Csm2, is enriched with Rad55 at autonomous replicating sequences, which contain replication origins

The Rad51 paralogs, Rad55 and Csm2, bind DNA and interact together to promote Rad51 filament formation (Gaines et al, 2015; Rosenbaum et al, 2019). To determine if either Rad55 or Csm2

binds to specific DNA sequences, we took an unbiased approach and performed chromatin immunoprecipitation sequencing (ChIP-seq) experiments. We immunoprecipitated C-terminal 9MYC-tagged Rad55 and 6HA-tagged Csm2 from asynchronous cultures. We verified that cells expressing these tagged proteins were functional by examining their MMS sensitivity by serial dilution (Fig. EV1A). In ChIP-seq analysis, we identified the binding sites of both Rad55 and Csm2 to specific DNA target regions (Fig. 1A). Rad55 binds to more DNA sites compared to Csm2 (Fig. 1A; 989 peaks vs 605 peaks, respectively). Csm2 DNA binding sites overlap significantly with Rad55 peaks (91% overlap) (Figs. 1A and EV1B; 553 of the 605 Csm2 total binding sites). Direct correlation of the binding peaks of Csm2 revealed a significant enrichment of Csm2 binding to ARS sites, which contain the replication origins in yeast (Brewer and Fangman, 1987) (Fig. 1B). In contrast, taking all Rad55 peaks into account those are not significantly enriched at ARS sites (Fig. EV1C). However, those Rad55 peaks that are also targeted by Csm2 strongly associate with ARS sites (Fig. 1C). Indicating that Csm2 and Rad55 may function together at ARS sites.

To further characterize the function of Csm2 and Rad55 at ARS sites, we investigated if the binding may correlate to DNA damage. Since Rad52 also physically interacts with Rad55 and Csm2 to promote HR (Godin et al, 2013; Gaines et al, 2015), we asked whether Rad52 is enriched with either Csm2 or Rad55 in the same DNA regions. By using publicly available ChIP-seq results for Rad52 (Costantino and Koshland, 2018), we revealed that indeed Rad52 DNA binding overlaps Csm2 and Rad55 binding peaks (Fig. EV1D,E). Further analysis revealed that Csm2, Rad55, and Rad52 all overlap at ARS (see for example 15 kb region on chromosome II, binding peaks of Csm2, Rad55, and Rad52 are plotted, they bind to genomic regions of the replication initiation factor MCM2-7 and the ARS site, ARS219.5) (Fig. EV1F). This overlap suggests that these proteins may jointly act at replication origins, potentially stabilizing these regions and supporting replication initiation processes. Consistent with a role in the bypass of replicative damage, we also find that Csm2 (Fig. 1D), and Rad55 (Fig. 1E) are significantly enriched at DNA regions where replication stalls/slow as indicated by high levels of DNA Polymerase 2 (DNA Pol2) occupancy ($p < 0.0001$). These findings indicate that Csm2 and Rad55 are significantly enriched at ARS sites and are associated with regions where replication is challenged, as evidenced by high DNA Polymerase 2 (DNA Pol2) occupancy ($p < 0.0001$), a marker of stalled or slowed replication. The overlap of Rad52, Csm2, and Rad55 binding at ARS regions, particularly in proximity to MCM2-7 and ARS219.5, suggests that these proteins may work in concert at replication origins to support genome stability, especially during replication stress.

The Shu complex plays a pivotal role in the tolerance of damage arising during replication (Ball et al, 2009). Therefore, we asked whether the protein levels of the Shu complex members (Csm2, Psy3, Shu1, and Shu2) are cell cycle regulated. To address this question, we first synchronized cells expressing the Shu complex primary DNA binding protein, Csm2-6HA, in G1 using α-factor. The cells were subsequently released into a fresh medium and protein levels of Csm2 were assessed over time by western blot (Fig. 1F). We find that Csm2 protein levels are present in G1 although largely reduced during G1 but increase during S/G2 phases, starting at 40 min after α-factor release and then reaching a plateau (Fig. 1F). In contrast, the G2/M cyclin, Clb2, protein levels

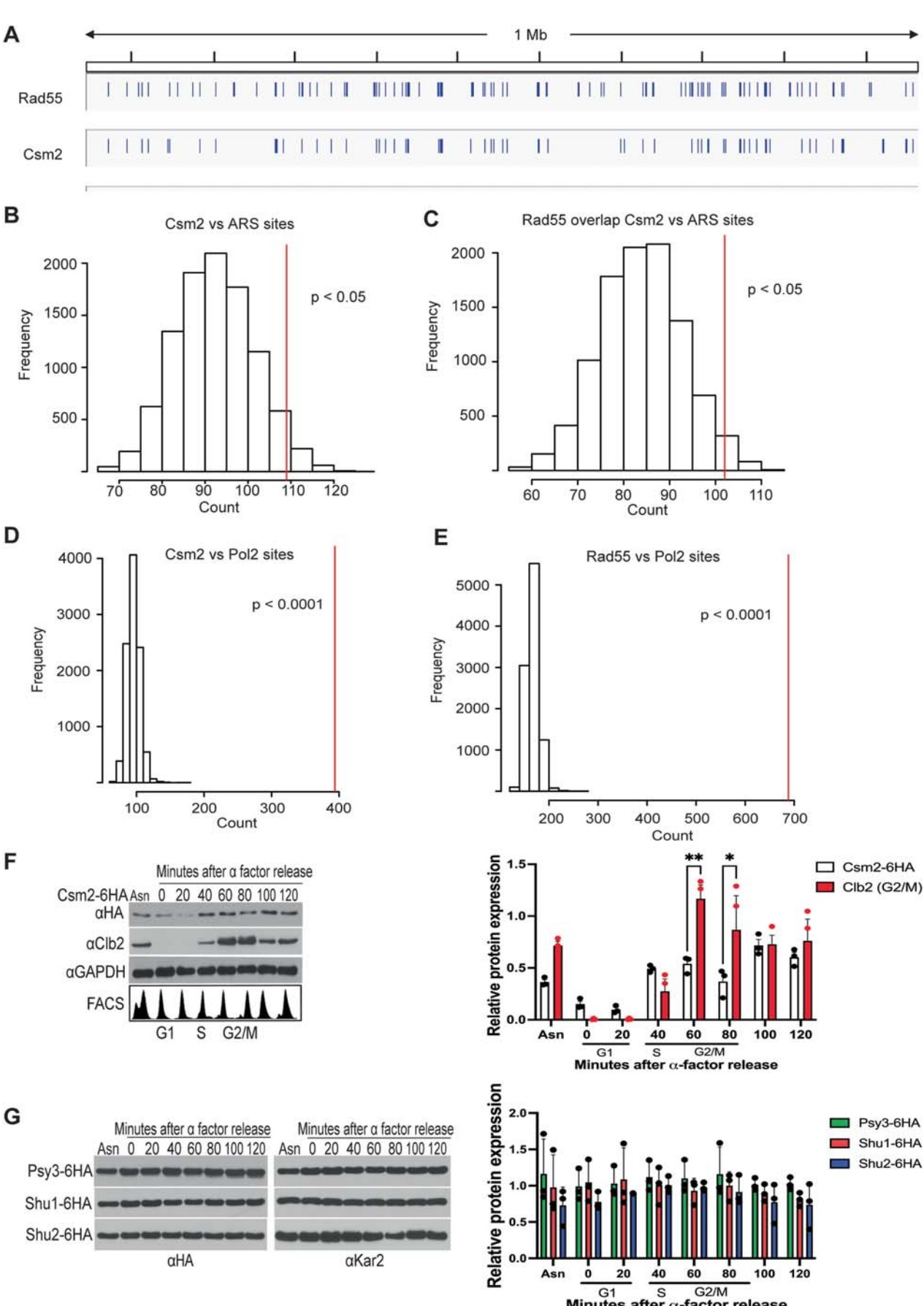

**Figure 1. Shu complex member, Csm2 is enriched at the replication origin and cell cycle regulated.**

(A) IGV Illustration of binding sites obtained by ChIPseq of Rad55 and Csm2 on Chromosome 7. Using the genome browser, all peaks obtained by ChIPseq were visualized and aligned to the yeast genome. As an example, we visualized the binding sites of Rad55 and Csm2 on Chromosome 7. (B) overlap of Csm2 to annotated ARS regions. Exact $p$-value: $p = 0.0346$. (C) Overlap of joint peaks of RAD55 and Csm2 with ARS sites. Exact $p$-value: $p = 0.0275$. (D) Overlap of Csm2 peaks with DNA Pol II binding sites. DNA Pol II sites are obtained from previous publications and indicate regions of replication fork stalls or slow-downs (Paeschke et al, 2011). High DNA Pol II binding reflects a long occupancy of the replisome at that site and can be correlated to replication pausing. Exact $p$-value: $p = <0.0001$. (E) Overlap of RAD55 peaks with Pol II binding sites with red lines depicting $p$-value threshold delineating overlap between Rad55 and Pol2 sites. Exact $p$-value: $p = <0.0001$. (B–E) Bioinformatics analyses demonstrating the overlap of genomic features with the ChIPseq peaks. Significance is assessed by a genome-wide permutation test. Bars represent the normal distribution of 10,000 random genomes. Based on ChIPseq peaks, we calculated the overlap of ChIPseq binding sites to other genomic features. $P$-values denote the statistical significance of the enrichment of experimental-determined peaks to random genome sets. The location of the red lines indicates the significance; the further to the right the line is, the lower the calculated $p$-value is. (F) Csm2 expression increases during S/G2 and then plateaus. Csm2-6HA expressing cells were either untreated (asynchronous, AS) or cell cycle arrested in G1 with α-factor. The α-factor arrested cells were subsequently released into fresh YPD medium (0 min) and grown for 120 min. Protein samples from the indicated time points were analyzed by western blot for Csm2 (anti-HA), the G2/M cyclin, Clb2, (anti-Clb2), or a loading control, GAPDH (anti-GAPDH). Quantification from three experiments and the mean was calculated from SEM. The cell cycle stage was analyzed FACS. Asterisks indicate statistical significance in comparison with Clb2 and Csm2 in 60 and 80 min under the same experimental conditions using a student's t-test. Statistical comparisons are indicated as $*p < 0.05$ and $**p < 0.01$; where (60 min: Csm2-6HA vs Clb2 $p = 0.004$ and 80 min: Csm2-6HA vs Clb2 exact $p = 0.0332$. (G) Psy3, Shu1, and Shu2 proteins are expressed throughout the cell cycle. Psy3, Shu1, and Shu2 6HA-expressing cells were either untreated (asynchronous, AS) or cell cycle arrested in G1 with α-factor. Protein samples from the indicated time points were analyzed by western blot for Psy3, Shu1, and Shu2 (anti-HA), the G2/M cyclin, Clb2, (anti-Clb2), or a loading control, Kar2 (anti-Kar2). Quantification from three experiments is shown. The error bar represents the mean ± SEM. Source data are available online for this figure.

peak at 60–80 min after α-factor release (Fig. 1F). Cell cycle progression was confirmed by FACS analysis (Fig. 1F). The elevated expression of Csm2 during S/G2 is consistent with a role during replication when HR is also active. Note that we previously reported that Csm2 protein levels were not cell cycle regulated (Godin et al, 2016b). However, upon closer inspection, we found that the blots were over-exposed and therefore the increased expression of Csm2 during S/G2 was not readily apparent. We next assessed whether the other components of the Shu complex (Psy3, Shu1, or Shu2) are expressed during specific cell cycle stages. Unlike the Csm2 protein, Psy3, Shu1, and Shu2 protein levels remain relatively constant throughout the time course [Fig. 1G; (Godin et al, 2016b)]. Overall, the increased expression of Csm2 during S/G2 phases suggests that the Shu complex is primed to function during replication and HR. However, this does not exclude its role in the earlier cell cycle stages, particularly G1/S, where Csm2, Rad55, and Rad52 binding were observed at ARS sites. The binding of Csm2 to ARS regions in G1/S may occur at relatively lower protein levels as observed in (Fig. 1F), sufficient to facilitate early-stage interactions that prepare replication origins for the upcoming demands of S phase, potentially stabilizing these regions before full replication begins. This biphasic engagement aligns with Csm2's role in genome stability, with low protein expression and binding in G1/S and elevated expression during S/G2 complementing one another to ensure faithful DNA replication and repair throughout the cell cycle.

## Csm2 enrichment to ARS sites depends on its DNA binding activity and interaction with Rad55

Next, we determined the conditions and genetic determinants of when Csm2 is recruited to ARS sites using chromatin immuno-precipitation (ChIP) experiments. To verify that Csm2 enrichment at ARS sites depends on its DNA binding activity, we performed ChIP experiments using a Csm2 DNA binding mutant (Csm2-K189A, R190A, R191A, R192A-6HA) (Rosenbaum et al, 2019). Indeed, without DNA binding activity Csm2 does not bind to three selected ARS sites, ARS216, ARS306, or ARS305 (Fig. 2A). As a negative control binding of Csm2-6HA and the Csm2 DNA binding mutant was monitored at an intergenic region on

Chromosome IV, which is 6.2 kb away from ARS412 and ARS413 (Fig. 2A; Chr IV IG).

The Shu complex functions to bypass alkylation-induced DNA damage during S phase. Therefore, we next determined whether Csm2 would still bind to ARS sites when cells are exposed to the alkylating agent, methyl methanesulfonate (MMS), which would stall/arrest the cells in S phase. We find that Csm2 enrichment at ARS216, ARS306, and ARS305 is reduced in MMS-exposed cells (Fig. 2B). Note, this reduction in binding is not due to overall changes in ChIP efficiency in these strains. We previously showed that Csm2-Psy3 preferentially binds and enables bypass of abasic sites, which are created upon removal of alkylated bases during base excision repair, during S phase (Rosenbaum et al, 2019). Therefore, we asked whether an accumulation of abasic sites by deletion of the AP endonucleases that process them (APN1 and APN2) would also result in decreased Csm2 enrichment to ARS sites. Similar to what we observed with the MMS treatment, when abasic sites accumulate, Csm2 enrichment to ARS sites is also reduced (Fig. 2C). One possibility is that Csm2 protein levels may be reduced upon MMS exposure. Therefore, we analyzed the steady-state protein levels of Csm2-6HA, as well as its binding partner Psy3-6HA, upon increasing MMS concentrations and overtime (Fig. EV2). We do not observe gross changes in either Csm2 or Psy3 protein levels upon MMS exposure (Fig. EV2). Therefore, loss of protein expression cannot explain why Csm2 enrichment at ARS sites is reduced. These findings suggest a model in which either Csm2 moves with the replication fork away from the replication origin during S phase or is relocated to alkylation-induced DNA damage at other sites.

Next, we asked whether Csm2 relocalizes away from the ARS sites when cells are arrested in S phase or when DSBs are induced. To do this, we used hydroxyurea treatment, which depletes dNTP pools or exposed the cells to bleomycin, which directly creates DSBs (Fig. EV3A,B). In both cases, Csm2 enrichment at ARS sites is significantly reduced (Fig. EV3A,B). In contrast to Csm2, the canonical Rad51 paralog, Rad55, enrichment to ARS sites is not significantly changed upon MMS treatment (Fig. EV3C). Similarly, the enrichment of the MCM complex member, Mcm4, at ARS sites is not significantly decreased upon HU or bleomycin exposure (Fig. EV3D,E). Therefore, these findings suggest that Csm2 loss at ARS sites upon DNA damage exposure is not a general feature of recombination proteins or replication factors.

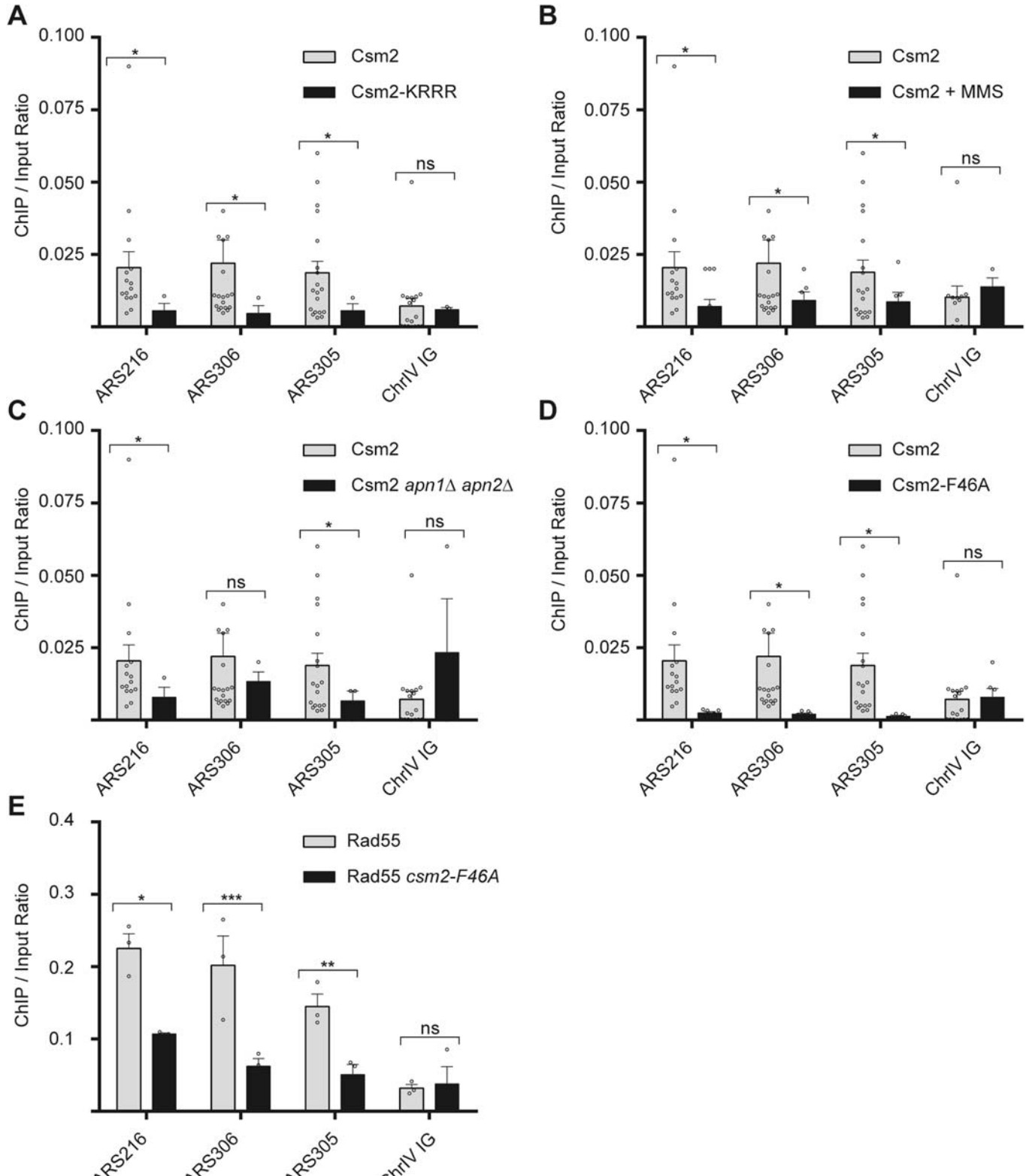

Csm2 directly interacts with Rad55 and together with the other Rad51 mediators promotes Rad51 presynaptic filament assembly (Gaines et al, 2015). Since Csm2 and Rad55 are enriched at ARS sites together, we then assessed whether the physical interaction between Csm2 and Rad55 is necessary for their enrichment at ARS

sites (Fig. 2D,E). To address this, we took advantage of a Csm2 mutant, Csm2-F46A, that maintains Shu complex integrity while disrupting the Csm2-Rad55 binding interface (Gaines et al, 2015). We find that the *Csm2-F46A-6HA* mutant exhibits greatly reduced binding to ARS sites (Fig. 2D). These results suggest that Csm2's

**Figure 2. Csm2 enrichment at ARS sites depends on DNA binding and its interaction with Rad55 and is reduced upon MMS damage or when abasic sites accumulate.**

(**A**) ChIP and qPCR of Csm2 and the DNA binding mutant, Csm2-KRRR. qPCRs were performed at three ARS sites (ARS216, ARS306, ARS305) and one control (ChrIV IG). Csm2 vs. Csm2KRRR ARS216 $p = 0.012298$, ARS305 $p = 0.049031$, ARS306 $p = 0.016581$, and ChrIV IG $p = 0.588828$. (**B**) ChIP and qPCR of Csm2 untreated or treated with 0.02% MMS. qPCRs were performed at three ARS sites (ARS216, ARS306, ARS305) and one control (ChrIV IG). Csm2 vs. Csm2+MMS ARS216 $p = 0.025306$, ARS305 $p = 0.019218$, ARS306 $p = 0.045902$, ChrIV IG $p = 0.213139$. (**C**) ChIP and qPCR of Csm2 and Csm2 *apn1Δ apn2Δ*. qPCRs were performed at three ARS sites (ARS216, ARS306, ARS305) and one control (ChrIV IG). Csm2 vs. Csm2 *apn1Δ apn2Δ* ARS216 $p = 0.013517$, ARS305 $p = 0.010209$, ARS306 $p = 0.066661$, ChrIV IG $p = 0.073803$. (**D**) ChIP and qPCR of Csm2 and *Csm2-F46A*. qPCRs were performed at three ARS sites (ARS216, ARS306, ARS305) and one control (ChrIV IG). Csm2 vs.*Csm2-F46A* ARS216 $p = 0.011423$, ARS305 $p = 0.016839$, ARS306 $p = 0.013847$, ChrIV IG $p = 0.097323$. (**E**) ChIP and qPCR of RAD55 and RAD55 *Csm2-F46A*. qPCRs were performed at three ARS sites (ARS216, ARS306, ARS305) and one control (ChrIV IG). RAD55 vs. RAD55 *Csm2-F46A* ARS216 $p = 0.017328$, ARS305 $p = 0.001765$, ARS306 $p = 0.000594$, ChrIV IG $p = 0.123547$. All plotted results were based on the average of the ChIP/Input ratio of at least three independent experiments ± SEM. Significance was calculated based on a one-sided Student's t-test. Asterisks indicate statistical significance in comparison with wild-type cells under the same experimental conditions. *$p < 0.05$, **$p < 0.01$, ***$p < 0.001$ and ns = not statistically significant. Source data are available online for this figure.

interaction with Rad55 is necessary for its efficient enrichment to ARS sites which is independent of its DNA binding activity. We next determined whether Rad55 would be recruited to ARS sites if the *csm2-F46A* mutation disrupted its interaction with Csm2 (Fig. 2E). Indeed, Rad55 enrichment to ARS sites is significantly reduced in the *csm2-F46A* mutant, albeit not as profound as what is observed for Csm2 (Fig. 2E). Together these results demonstrate that the Shu complex interaction with Rad55 is necessary for efficient enrichment of both factors to ARS.

## The Shu complex physically interacts and co-evolves with the ORC and MCM complexes

Many replication initiation factors bind to ARS sites to enable efficient DNA replication during S phase. Since we find that Csm2 is enriched at ARS sites, it is possible that the Shu complex may physically interact with other proteins that are also found at replication origins. Briefly, replication initiation begins with the binding of the ORC complex to replication origins in G1. Subsequently, the MCM complex is recruited during G1 and activated through phosphorylation during S phase to form the CMG helicase, consisting of Cdc45, the MCM complex, and the GINS (Psf1-3 and Sld5) (Bochman and Schwacha, 2009). In addition to these factors, the MTC, composed of checkpoint proteins namely Mrc1–Tof1–Csm3, is also important during S phase to improve the efficiency and protection of the replisome (Lewis et al, 2017). Since there are many factors critical for replication initiation, we used evolutionary rate covariation (ERC) analyses to identify key complexes that may participate in a genetic or physical interaction with the Shu complex. ERC analysis examines how the evolutionary rate of each protein changes between species (in this case between 343 yeast species) and measures the correlations between proteins concerning these species-specific rates. It has been shown that protein pairs with high ERC scores, i.e., those that are strongly positively correlated, tend to be functionally related (Clark et al, 2012; Priedigkeit et al, 2015). Their rates correlate because co-functional proteins experience the same selective pressures over time, and hence their evolutionary rates respond to those pressures in parallel, leading to simultaneous accelerations or decelerations of amino acid changes (Clark et al, 2012; Little et al, 2024). Therefore, ERC analysis is valuable for identifying previously unrecognized genetic or physical interactions.

We have successfully used this approach to identify novel DNA repair factors and to find new protein interactions (Clark et al, 2012; Godin et al, 2015; Huang et al, 2020; Brunette et al, 2019; Böhm et al, 2016). For this study, we used a dataset of 343 yeast species to calculate the ERC values between all members of the Shu complex for their correlations with other replication-related complexes. We asked whether the Shu complex proteins (Shu1, Shu2, Csm2, or Psy3) positively correlate with members of the CMG, GINS, MCM, ORC, or MTC complexes. We find that the Shu complex is highly and significantly correlated with the CMG helicase, the MCM, and ORC complexes but not with the GINS or MTC (Fig. 3A; each dot represents a unique protein-protein pair evolutionary rate covariation). Statistical significance was assessed by comparing Shu complex proteins to random gene sets drawn from the genome, for which the expectation is that the correlation is zero (permutation test). Like yeast, the human Shu complex is also significantly positively correlated with the same complexes (Fig. 3B; CMG helicase, MCM, and ORC complexes). This was determined from a dataset of 63 mammal species. We find that ERC values between human Shu complex proteins also correlated more with members of these 3 complexes than random gene sets (permutation test). These results suggest that the yeast and human Shu complex co-evolve with specific replication initiation factors.

Co-evolution between two proteins could be due to genetic or physical interaction. Therefore, we asked whether the Shu complex or the canonical Rad51 paralogs, Rad55-Rad57, interact with members of the MCM or ORC complexes. By using a yeast-2-hybrid (Y2H) approach, we can quickly assess protein interactions where the Shu complex member Shu1, Shu2, Csm2, or Psy3 is cloned into a plasmid expressing the GAL4-DNA binding domain (pGBD) and the MCM complex member Mcm2-7 is cloned into a GAL4-DNA activating domain (pGAD) expressing plasmid. If two proteins interact, then a Y2H interaction would be detected by plating the cells on a selective medium that relies on the transcription of a downstream reporter gene, histidine, for growth. By Y2H, we find that Shu2, Csm2, and Psy3 interact with multiple MCM complex members, whereas Rad55 and Rad57 do not (Fig. 3C; top panels). Note that Psy3 interacts with all MCM complex members by Y2H, whereas Shu2 or Csm2 interacts with a subset of MCM proteins (Fig. 3C). Mcm4 is a positive control, and the empty vectors are negative controls.

We then assessed if the Shu complex interacted with the ORC complex members by Y2H. Similar to what we observed with the MCM complex, Shu2, Csm2, and Psy3 exhibit a Y2H interaction with multiple ORC complex members (Fig. 3D; top panels). Psy3 interacts with all ORC complex members, whereas Shu2 and Csm2 interact with a subset of ORC complex proteins (Fig. 3D). We also observe a weak but reproducible Y2H interaction between Rad55 with Orc6 (Fig. 3D). Further analysis in (Fig. 5C) below, indicated

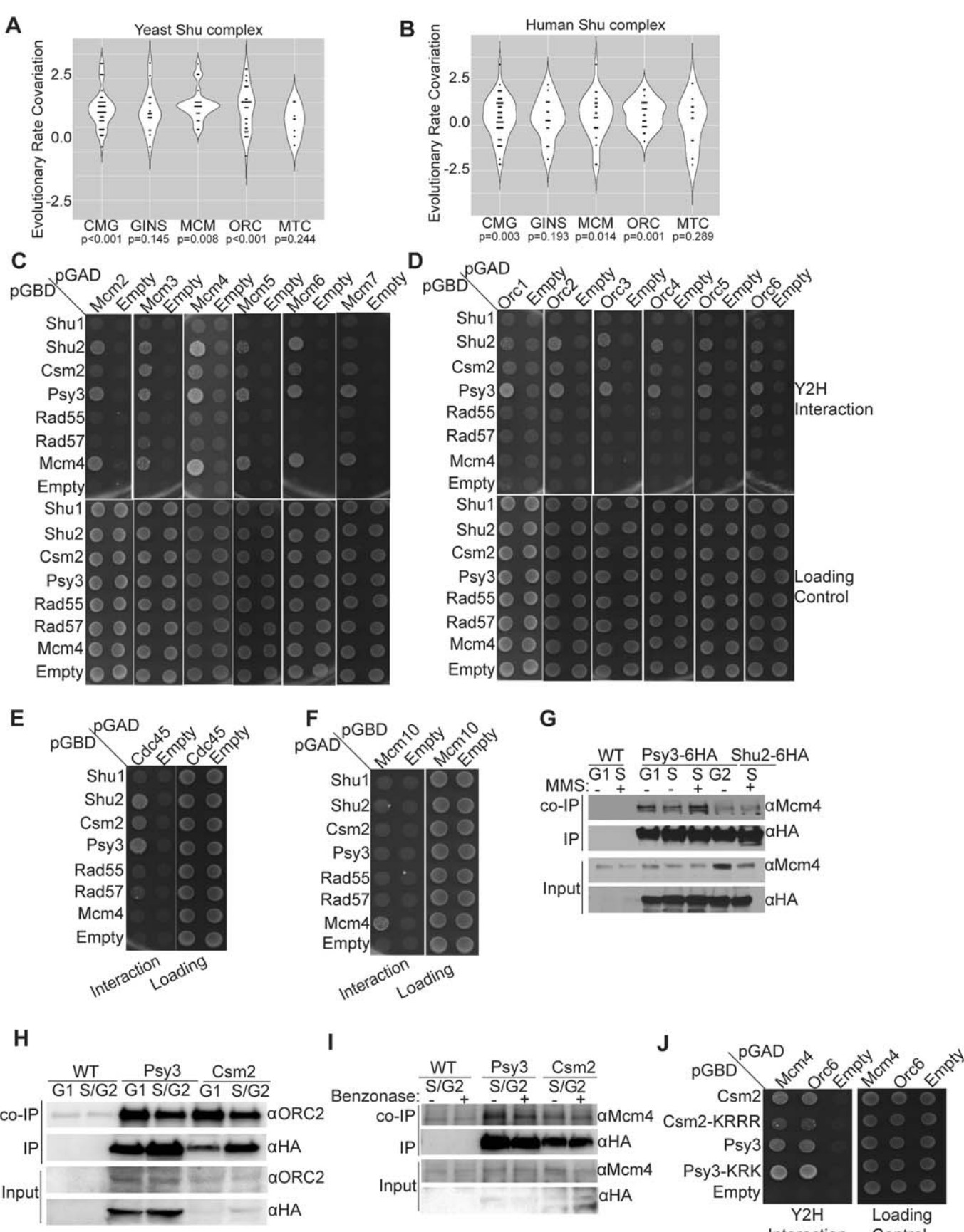

**Figure 3.  The Shu complex co-evolves and physically interacts with the ORC and MCM complex members.**

(A) The yeast Shu complex has evolutionary rates that correlate with CMG ($n = 44$, $p < 0.001$), MCM ($n = 24$, $p = 0.008$), and ORC ($n = 24$, $p < 0.001$) complexes at levels higher than expected by chance when compared to a null distribution of 1000 permutations. Sample sizes for non-significant comparisons were GINS $n = 16$, MTC $n = 12$, and RAD51 $n = 12$. Violin plots show higher evolutionary rate correlations between the yeast Shu complex and the replication initiation complexes when compared to the genome-wide background correlation, which is expected to be zero. Each dot represents co-evolution between two proteins compared to 1000 permuted nulls, the higher the correlation coefficient, the more significant the co-evolution. Significance is assessed by genome-wide permutation test contrasting observed Shu correlations against random protein sets. (B) Same as (A), except the human Shu complex was analyzed against human protein complexes. Violin plots show the same complexes CMG ($N = 48$, $p = 0.003$), MCM ($N = 24$, $p = 0.014$), and ORC ($N = 24$, $p = 0.001$) have significantly high ERC compared to a null distribution of 1000 permutations. (C) Shu complex members Shu2, Csm2, and Psy3 exhibit a Y2H interaction with Mcm complex members (MCM2-7). Y2H experiment examining Shu complex (Shu1, Shu2, Csm2, Psy3) interaction with MCM complex members compared to the canonical Rad51 paralogs, Rad55-Rad57, and MCM complex member Mcm4. (D) Shu2 and Psy3 exhibit a Y2H interaction with ORC complex members (ORC1-6). Y2H experiment examining Shu complex interaction with ORC complex members compared to Rad55-Rad57 and MCM complex member Mcm4. (E) Shu complex members Shu2, Csm2, and Psy3 exhibit a Y2H interaction with Cdc45. Y2H experiment examining Shu complex interaction with Cdc45 compared to Rad55-Rad57 and Mcm4. (F) Mcm10 does not exhibit a Y2H interaction with the Shu complex. Y2H experiment examining Shu complex interaction with Mcm10 compared to Rad55-Rad57 and MCM complex member Mcm4. For the Y2H experiments, yeast with the indicated plasmids were grown in SC-L-W, plated on SC-L-W-H medium, and incubated for 2 days at 30 °C. Growth on SC-L-W-H is indicative of a Y2H interaction, and SC-L-W is used as a loading control. Empty vectors are negative controls, and Mcm4 is a positive control. All experiments were done in triplicate. (G) Psy3 co-IPs with Mcm4 predominantly during the G1/S phase with or without MMS. Untagged wild-type, Psy3-6HA, or Shu2-6HA expressing cells were either arrested in G1, and S phase cells released from α-factor arrest ($+/-0.03\%$ MMS for 40 min) or arrested in G2/M with nocodazole. Psy3 is IP using HA antibodies and then runs on an SDS-PAGE gel. (H) Psy3 and Csm2 co-IP ORC2 during G1 and S/G2 phase cells. Untagged wild-type, Psy3-6HA, or Csm2-6HA expressing cells were arrested in G1 with α-factor or released from α-factor into S/G2. Psy3 and Csm2 was IP using HA antibodies and then run on an SDS-PAGE gel. Co-IP with ORC2 was accessed using anti-ORC2 antibodies, and immunoprecipitation was accessed using anti-HA antibodies. The input represents 5% of the total. (I) Co-IP experiments of the parental untagged strain, Psy3-6HA, and Csm2-6HA in the presence or absence of benzonase. Cells expressing Psy3-6HA, Csm2-6HA, or the untagged strain were arrested in G1 with α-factor and then released into fresh YPD medium for ~45 min, which correlated with S/G2 cell cycle phase. S phase was verified by the budding index. Psy3 or Csm2 were immunoprecipitated with αHA antibodies in the presence or absence of benzonase (25 U) and the protein was run on an SDS-PAGE gel and blotted for Mcm4 (αMcm4; co-IP) or HA (αHA; IP). (J) Csm2 and Psy3 Y2H interaction with Mcm4 and Orc6 are DNA binding independent. Y2H analysis of Csm2, Psy3, and the DNA binding mutants, Csm2-KRRR and Psy3-KRK with Mcm4 and Orc6. Yeast with the indicated plasmids were grown in SC-L-W, plated on SC-L-W-H medium, and incubated for 2 days at 30 °C. Source data are available online for this figure.

that the loss of PSY3, but not CSM2, may lead to a modest reduction in the interaction between Rad55 and Orc6. However, no additional Y2H interactions were observed between Rad55-Rad57 and the ORC complex members. In addition to the MCM and ORC complexes, Shu2, Csm2, and Psy3 also exhibit a Y2H interaction Cdc45, which is an integral member of the CMG helicase (Fig. 3E). However, the Shu complex does not interact with all replication initiation factors since we do not observe a Y2H interaction between the Shu complex and Mcm10, which interacts with the MCM complex (Fig. 3F).

To validate the physical interaction between the Shu complex and MCM and ORC complexes in cells, we performed co-IP experiments. In these assays, we immunoprecipitated Psy3-6HA or Shu2-6HA and blotted for endogenous Mcm4 (Fig. 3G). We arrested the cells in G1 using α-factor or released the cells into a fresh medium (with or without MMS) and performed the co-IP in S or G2/M cultures. The cell cycle stage was verified by FACS analysis. We find that Psy3 co-IPs Mcm4 in G1 as well as S phase cells in the presence or absence of MMS (Fig. 3G). In contrast, we observe a weak interaction between Psy3 with Mcm4 in G2 cells or with Shu2 in MMS-exposed S phase cells (Fig. 3G). These results suggest that Psy3 interacts with Mcm4 both in G1 and S phase in a DNA damage-independent manner. We next assessed whether Psy3 or Csm2 co-IP endogenous Orc2 in G1 when the ORC complex is loaded onto the replication origins or in S/G2 arrested cells (Fig. 3H). While Psy3 and Csm2 co-IP Orc2 in both G1 and S/G2, Psy3 and Csm2 pull down more Orc2 in the G1 arrested cells. Suggesting that these protein interactions are largely DNA independent, we find that Psy3 and Csm2 maintain their interaction with Mcm4 in the presence of nuclease benzonase (Fig. 3I). Similarly, Csm2-Psy3 DNA binding mutants, Csm2-KRRR and Psy3-KRK respectively, exhibit a Y2H interaction with both Mcm4 and Orc6 (Fig. 3J). Note that Csm2-KRRR Y2H

interaction is modestly reduced, whereas Psy3-KRK is not. In both the Y2H and co-IP experiments, Psy3 interaction with the MCM and ORC complexes is the most robust (Fig. 3). Together, these results suggest that the Shu complex interacts with the ORC and MCM complexes during G1 and then maintains an interaction with the MCM complex and the CMG helicase during S phase.

## The Shu complex interaction with the MCM and ORC complexes is independent of *RAD51* and *RAD55* but dependent on Shu complex member, *PSY3*

Since the Shu complex interacts with recombinase Rad51 and its paralog Rad55 (Godin et al, 2013; Gaines et al, 2015), we asked whether the loss of *RAD51* or *RAD55* would alter the interaction between the Shu complex with ORC or MCM complex members. To assess this, we knocked out either *RAD51* or *RAD55* in the Y2H strain and examined the interaction between the Shu complex with either Orc1, Orc2, Mcm4, or Mcm7 by Y2H (Figs. 4A,B and EV4A). Suggesting that the Shu complex interaction with the ORC and MCM complex is independent of *RAD51* and *RAD55*, we find that Shu2, Csm2, and Psy3 interaction with ORC and MCM complex members is unchanged in *rad51Δ* and *rad55Δ* cells (Fig. 4A,B). Consistently, we also find that loss of *RAD51* or the Rad51 loader, *RAD52*, does not significantly alter Csm2 enrichment at ARS sites (Fig. 4C,D). Therefore, the Shu complex interaction with the ORC or MCM complex members is independent of *RAD51* or *RAD55*.

Next, we asked if the interaction between the Shu complex and the MCM or ORC complexes may be dependent on one of the Shu complex subunits. We individually knock out the four Shu complex members in the Y2H strain and found that disruption of *PSY3* results in loss of Shu2 and Csm2 interaction with either Mcm4 or Orc6 (Fig. 4E). Consistently, Csm2 enrichment at ARS sites is also reduced in *psy3Δ* cells (Fig. 4F). Note that loss of *PSY3* results in

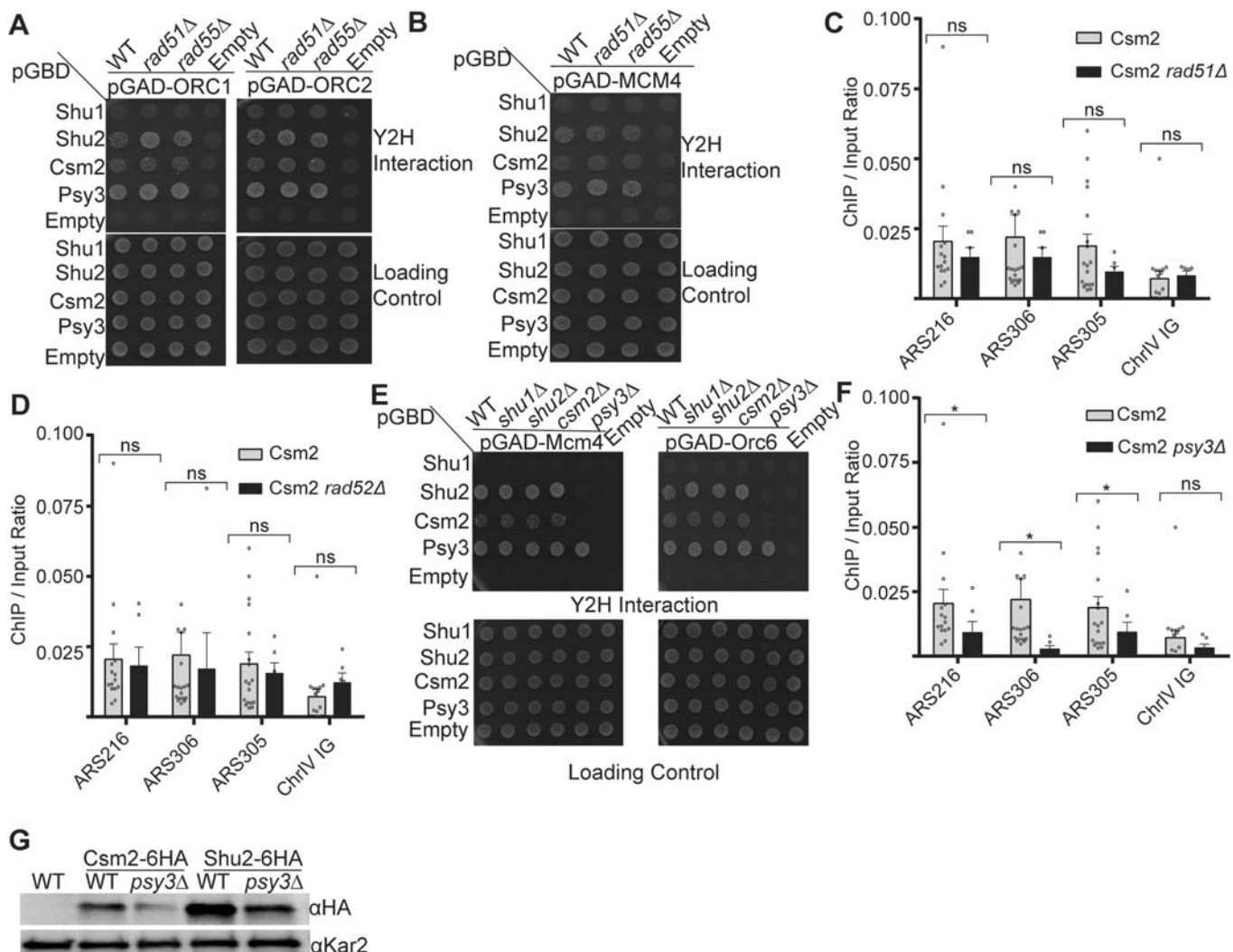

**Figure 4. The Shu complex interaction with the ORC and MCM complex is independent of Rad51, Rad52, and Rad55 but dependent on Psy3.**

(A) Shu complex Y2H interaction with ORC complex members (ORC1 and ORC2) is independent of *RAD51* and *RAD55*. Y2H experiment examining Shu complex (Shu1, Shu2, Csm2, Psy3) interaction with ORC1 and ORC2 transformed into Y2H strain with either *rad51Δ* or *rad55Δ* knocked out. (B) Same as (A) except the Mcm4 in a pGAD vector was analyzed. (C) ChIP and qPCR of Csm2 and Csm2 *rad51Δ*. qPCRs were performed at three ARS sites (ARS302, ARS306, ARS305) and one control (ChrIV IG) Csm2 vs Csm2 *rad51Δ* ARS216 $p = 0.464805$, ARS305 $p = 0.394199$, ARS306 $p = 0.331189$, ChrIV IG $p = 0.039181$. (D) ChIP and qPCR of Csm2 and Csm2 *rad52Δ*. qPCRs were performed at three ARS sites (ARS302, ARS306, ARS305) and one control (ChrIV IG). Csm2 vs Csm2 *rad52Δ* ARS216 $p = 0.303514$, ARS305 $p = 0.313981$, ARS306 $p = 0.230426$, ChrIV IG $p = 0.201505$. (E) Shu2 and Csm2 interaction with Mcm4 and Orc6 is dependent on *PSY3*. Y2H experiment examining Shu complex interaction with Mcm4 and Orc6 when each of the Shu complex members are deleted in the Y2H strain (*shu1Δ*, *shu2Δ*, *csm2Δ*, *psy3Δ*). For the Y2H experiments, yeast with the indicated plasmids were grown in SC-L-W, plated on SC-L-W-H medium, and incubated for 2–3 days at 30 °C. Growth on SC-L-W-H is indicative of a Y2H interaction and SC-L-W is used as a loading control. All experiments were done in triplicate. (F) ChIP and qPCR of Csm2 and Csm2 *psy3Δ*. qPCRs were performed at three ARS sites (ARS302, ARS306, ARS305) and one control (ChrIV IG). Csm2 vs Csm2 *psy3Δ* ARS216 $p = 0.017880$, ARS305 $p = 0.016139$, ARS306 $p = 0.019552$ and ChrIV IG $p = 0.053302$. For the ChIP experiments, all plotted results were based on the average of the ChIP/Input ratio of at least three independent experiments ± SEM. Significance was calculated based on a one-sided Student's t-test. Asterisks indicate statistical significance in comparison with wild-type cells under the same experimental conditions. *$p < 0.05$ and ns = not statistically significant. (G) Western blot of Csm2 and Shu2 protein levels in WT and *psy3Δ* cells. Protein was isolated by TCA precipitation from untagged, WT or *psy3Δ* cells expressing 6HA-tagged Csm2 or Shu2. Csm2 and Shu2 protein levels were assessed by western blot (αHA) or for equal protein loading (αKar2). Source data are available online for this figure.

reduced expression of Csm2 and Shu2 (Fig. 4G). Together, these findings indicate that while the Shu complex interacts with ORC and MCM complexes independently of *RAD51* and *RAD55*, the presence of Psy3 is essential for these interactions. Psy3 not only facilitates the physical association of Shu2 and Csm2 with the ORC and MCM complexes but also supports their stability and localization to ARS sites.

## Rad51 and Rad52 interact predominantly with Mcm4, and this interaction is dependent on Shu complex member, *PSY3*

The Shu complex interacts with Rad52 and Rad51 to enable the bypass of replication damage by template switch (Gaines et al, 2015). Therefore, we wondered if Rad51 or Rad52 would also

broadly interact with members of the MCM or ORC complexes (Fig. 5A). By Y2H analysis, we find that Rad51 and Rad52 exhibit a Y2H interaction with Mcm4 and Rad51 weakly interacts with Mcm5 (Fig. 5A). However, we do not observe an interaction between Rad51 or Rad52 with Mcm2 and Mcm7 or ORC complex members (Orc2, Orc3, Orc4 and Orc6) suggesting that Rad51 and Rad52 predominantly interact with MCM complex members, Mcm4, and not the ORC complex (Fig. 5A). These results are consistent with previous findings demonstrating that Rad51 and Rad52 coIP Mcm4 (Cabello-Lobato et al, 2021).

Since the Shu complex interaction with the MCM and ORC complexes is likely dependent on Psy3, we asked whether Rad51 or Rad52 interaction with Mcm4 would similarly be *PSY3* dependent. To address this, we performed Y2H experiments between Rad51 or Rad52 with Mcm4 in a *psy3Δ* strain and found that Rad51 and Rad52 Y2H interaction with Mcm4 is Psy3 dependent (Fig. 5B). Since we find in Fig. 3D that Orc6 exhibits a Y2H interaction with Rad55, we examined whether Rad55 and Orc6 interaction depends on the Shu complex members, Csm2 or Psy3. Our result suggests that the loss of *PSY3*, but not *CSM2*, may modestly reduce the interaction between Rad55 and Orc6 (Fig. 5C), suggesting that Psy3 plays a subtle but important role in facilitating the Rad55-Orc6 interaction. Together, these results highlight a specific and functionally relevant interaction network between Rad51, Rad52, and the MCM complex, predominantly via Mcm4. Moreover, these interactions are tightly regulated by Psy3, underscoring its critical role in linking the Shu complex with the MCM and ORC complexes. This dependency on Psy3 may reflect a broader mechanism by which the Shu complex coordinates template switching and replication progression, further emphasizing the significance of Psy3 in replication stress responses.

## The N-terminus of Psy3 interacts with the ORC and MCM complex and is necessary and sufficient for error-free bypass of MMS-induced DNA damage

We identified a novel interaction between the Shu, MCM, and ORC complexes. Therefore, we wanted to address whether these physical interactions are critical for the Shu complex function to bypass MMS-induced DNA damage. To identify where the Shu complex may interact with MCM and ORC complexes, we used Alpha-fold to map the protein interfaces. Unfortunately, we do not observe any consistent models for these interactions with either the Shu complex or Rad51 and Rad52 with MCM or ORC complexes. Since the structure of the Shu complex was previously solved (Sasanuma et al, 2013; Zhang et al, 2017; Tao et al, 2012; She et al, 2012), we examined solvent-exposed residues outside the Shu complex binding interfaces that are highly conserved amongst fungi in Psy3 (Fig. EV5). We find that the N-terminus of Psy3 is both solvent-exposed and well-conserved (Fig. 6A,B). By mutating the N-terminal residues in Psy3 to alanine, we identified multiple mutants that maintain Psy3 Y2H interactions with the other Shu complex members, Csm2 and Shu1, but are deficient in their Y2H interactions with Mcm4 and Orc6 (Fig. 6C; *Y10A, L12A, F15A, T17A*, and *S18A*). These results suggest that the N-terminus of Psy3 is necessary and sufficient for its interactions with Mcm4 and Orc6. To determine if the interaction between Psy3 and the replication machinery is important for the Shu complex function, we examined whether these N-terminal residues are needed for MMS resistance. By complementing a *psy3Δ* cell with a

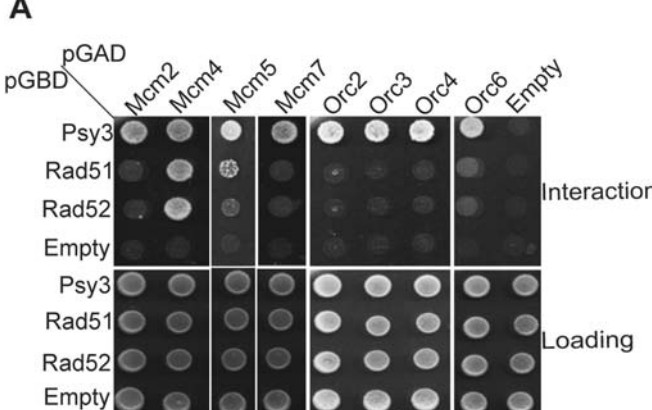

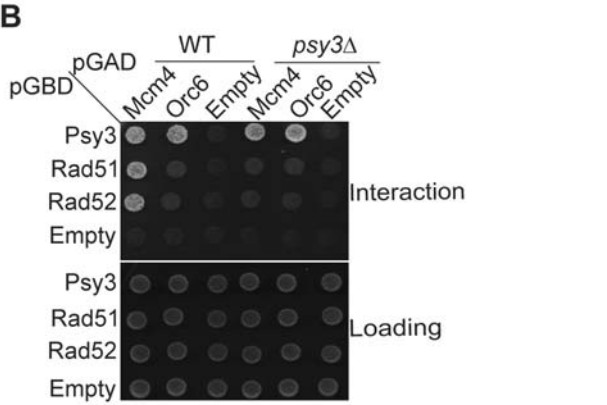

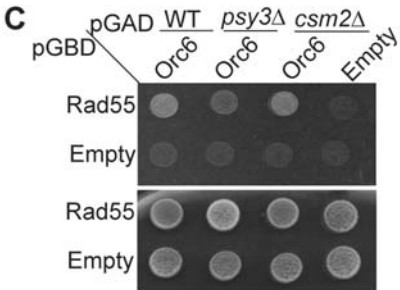

**Figure 5. Rad51 and Rad52 exhibit a strong Y2H interaction with Mcm4 that is dependent on the Shu complex protein, Psy3.**

(A) Rad51 and Rad52 exhibit a strong Y2H interaction with Mcm4 and a weak interaction with Mcm5 but not Mcm2, Orc4, or Orc6. (B) Rad51 and Rad52 Y2H interaction with Mcm4 is Psy3-dependent. Y2H analysis of Mcm4, and Orc6, with Rad51 and Rad52 transformed in either WT or in a *psy3Δ* yeast cell (PJ69-4α). (C) Rad55 Y2H interaction with Orc6 is modestly reduced upon *PSY3* deletion. The Y2H experiment examining Rad55 interaction with Orc6 transformed into Y2H strain with either *psy3Δ* or *csm2Δ* knocked out. Yeast with the indicated plasmids were grown in SC-L-W, plated on SC-L-W-H medium, and incubated for 2–3 days at 30 °C. All experiments were done in triplicate. Source data are available online for this figure.

plasmid expressing either *PSY3-Y10A, -L12A, -F15A, -T17A*, or *-S18A*, we find that all these mutants are unable to fully complement a *psy3Δ* cell MMS sensitivity in comparison to WT *PSY3* (Fig. 6D). We performed western blot to confirm that these N-terminal *psy3* mutants are expressed (Appendix Fig. S1A). Note that *PSY3-Y10A* is in an invariant residue and confers the greatest MMS sensitivity

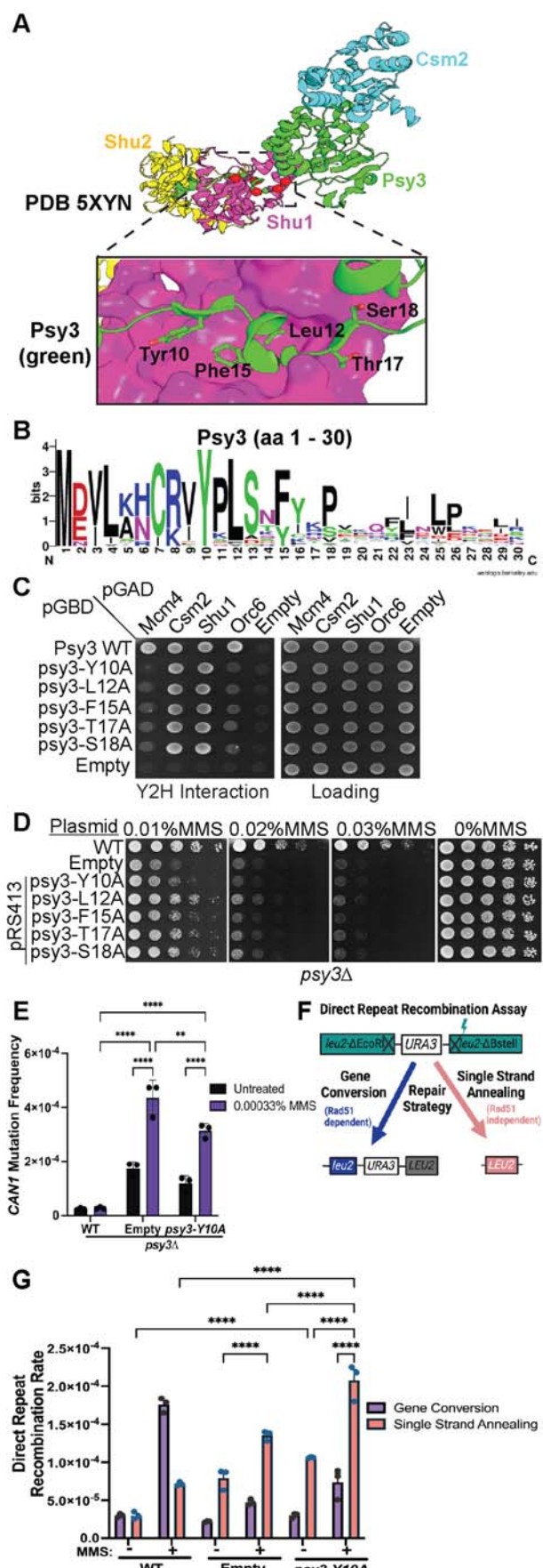

**Figure 6. The conserved N-terminus of Psy3 mediates Shu complex interaction with Mcm4 and Orc6.**

(A) The structure of the Shu complex (Shu1- pink, Shu2-yellow, Csm2-blue, Psy3-green) PDB: 5XYN, where Psy3 solvent-exposed residues outside of the Shu complex protein interactions, are highlighted (Robert and Gouet, 2014). (B) Weblogo showing the conservation of the N-terminus of Psy3, amino acids 1–30 (Crooks et al, 2004). (C) Mutation in the N-terminus of Psy3 results in loss of Mcm4 and Orc6 Y2H interactions but does not disrupt its interactions with the other Shu complex members, Csm2 and Shu1. Psy3 wild-type (WT) or the indicated psy3 mutant with either Mcm4, Csm2, Shu1, or Orc6 was transformed into the Y2H strain (PJ69-4α). All experiments were done in triplicate. (D) Psy3 N-terminus mutants are sensitive to MMS. *PSY3* disrupted cells were transformed with a wild-type *PSY3* expressing plasmid or a plasmid containing the indicated *PSY3* N-terminal mutants. Equal cell numbers of the indicated strains were five-fold serial diluted onto rich medium (YPD; 0% MMS) or rich medium with MMS (0.01%, 0.02%, or 0.03%). The plates were grown at 30 °C for 2 days and photographed. The experiment was performed in triplicate. (E) Spontaneous and MMS-induced mutation rates at the *CAN1* locus were measured in WT, empty, and *psy3-Y10A*, transformed in *psy3Δ*. *Psy3–Y10A* expressing cells exhibit an elevated mutation rate in a canavanine mutagenesis assay. Exact *p* values (empty: untreated vs. treated *p* = <0.0001, *psy3-Y10A*: untreated vs. treated *p* = <0.000, treated: empty vs. *psy3-Y10A p* = 0.0015, treated: WT vs. empty *p* = <0.0001, and treated: WT vs. *psy3-Y10A p* = <0.0001). The mean value and SD from three independent experiments are plotted as error bars. Significance was determined by Tukey's multiple comparisons test, where ** represents *p* < 0.0015 and **** represents *p* < 0.0001. (F) Schematic of direct repeat recombination (DRR). This assay involves two disrupted *leu2* genes, each containing a distinct restriction site, *EcoR1* and *BstEII* separated by an intervening *URA3* gene. Reversion to *LEU+* can be achieved via two distinct pathways. The Rad51-dependent gene conversion (GC) pathway leads to the restoration of *LEU+* alongside the retention of the *URA3* marker, producing *URA3 + LEU2+* recombinants. Alternatively, the Rad51-independent single-strand annealing (SSA) pathway restores *LEU+* while resulting in the loss of the *URA3* gene, yielding *URA3- LEU2+* recombinants. The DRR assay, therefore, provides a precise measure of recombination events by distinguishing between these mechanistically distinct repair pathways. (G) WT, *psy3Δ* or *psy3-Y10A* expressing cells harboring a direct repeat HR reporter (*leu2-ΔEcoRI::URA3::leu2-ΔBstEII*) were tested for spontaneous rates of GC and SSA and upon 0.0003% MMS exposure as described. Exact *p* values (SSA: empty-MMS vs. empty+MMS *p* =≤ 0.0001, GC vs. SSA: *psy3 Y10A + MMS p* =≤ 0.0001, SSA: WT-MMS vs. *psy3-Y10A-MMS p* = ≤0.0001, SSA: *psy3-Y10A-MMS* vs. *psy3-Y10A + MMS p* = ≤0.0001, SSA: empty+MMS vs. *psy3-Y10A + MMS p* = ≤0.0001, SSA: WT + MMS vs. *psy3-Y10A + MMS p* = ≤0.0001). Nine independent colonies were measured for each experiment and the mean value and SEM from three experiments (horizontal bar) were plotted. Significance was determined by the Tukey test where **** represents *p* ≤ 0.0001. Source data are available online for this figure.

(Fig. 6D). Therefore, we used this mutant as an N-terminal representative for additional studies.

The Shu complex prevents translesion synthesis-induced mutations (Shor et al, 2005; Ball et al, 2009), therefore we asked whether mutations would be increased in *psy3Δ* cells expressing *psy3-Y10A*. To do this, we performed a *CAN1* mutagenesis assay where mutations in the arginine permease gene, *CAN1*, transport a toxic arginine analog, canavanine, that is added to the medium (Whelan et al, 1979). Mutations in *CAN1* enable the cells to survive canavanine exposure. By measuring the number of colonies with a mutation in *CAN1*, we can calculate the cellular mutation frequency. We find that *psy3-Y10A* expressing cells exhibit a significant increase in spontaneous and MMS-induced mutations relative to WT that are comparable to a *psy3Δ* cell (Fig. 6E). Note that upon MMS treatment, *psy3Δ* cells exhibit more mutations relative to a *psy3-Y10A* expressing cell. These results suggest that Psy3 interaction with the replication machinery is important for preventing spontaneous and MMS-induced mutations.

In addition to Shu complex function to prevent mutations, it also plays a critical role in promoting Rad51-mediated gene conversion (Godin et al, 2013; Gaines et al, 2015; Godin et al, 2016b; Rosenbaum et al, 2019). To examine whether loss of the Shu complex interaction with the replication machinery results in changes in recombination, we employed a direct repeat recombination assay (*leu2-ΔEcoRI::URA3::leu2-ΔBstEII*). This assay enables quantification of overall recombination rates and allows us to distinguish between different recombination mechanisms. As illustrated in Fig. 6F, recombination events resulting in a functional *LEU2* gene can occur via Rad51-dependent gene conversion (GC), which preserves the intervening *URA3* gene, or through Rad51-independent single-strand annealing (SSA), which deletes the intervening *URA3* gene. SSA occurs with extensive resection that exposes homology between the two *LEU2* sequences. However, it has been demonstrated in similar assays that SSA-like outcomes can also arise through a Rad51-independent, inter-sister chromatid recombination event (Dong and Fasullo, 2003). We examined whether *psy3-Y10A* expressing cells result in fewer Rad51-mediated GC events, both spontaneously and upon MMS damage when compared to WT *PSY3* (Fig. 6G). We find that psy3-Y10A results in fewer GC events upon MMS damage (Fig. 6G; $p ≤ 0.0001$). Furthermore, spontaneous recombination leads to an approximately equal number of GC and SSA events in WT cells (Fig. 6G). Upon MMS damage, the number of GC events is increased relative to SSA (Fig. 6G). In contrast, both *psy3Δ* and *psy3-Y10A* expressing cells exhibit more SSA events relative to GC, even in the presence of MMS damage, despite overall recombination rates being increased (Fig. 6G; Appendix Fig. S1B). Therefore, only the MMS-treated WT cells displayed a pronounced preference for Rad51-mediated GC, which is dependent on Psy3's interaction with the MCM and ORC complexes. These findings collectively suggest that the conserved N-terminus of Psy3 is necessary and sufficient for the error-free bypass of MMS-induced DNA damage.

## Discussion

To understand how the Shu complex facilitates accurate DNA damage bypass during S phase, we investigated the binding patterns of Csm2, the primary DNA binding subunit within the Shu complex, in *S. cerevisiae*. Our observations revealed that Csm2 is notably enriched at ARS sites and is evolving in a correlated pattern with several replication initiation complexes including the ORC and MCM complexes. Moreover, multiple members of the Shu complex interact with the ORC complex through Y2H and co-IP, notably during G1 phase and independently of DNA. Subsequently, the Shu complex engages with the MCM complex and maintains this association in S phase. Demonstrating that Psy3 is critical for mediating the Shu complex interaction with the MCM and ORC complexes, we also observe that the Shu complex interaction with the replication machinery is depleted in *psy3Δ* cells. Through mapping the interaction interface between Psy3 and the ORC and MCM complexes, we identified residues in the N-terminus of Psy3 that are necessary and sufficient for these protein interactions. Mutating the conserved *Y10* residue in Psy3 results in MMS sensitivity, increased mutations, and changes in recombination-mediated gene conversion upon MMS, thus indicating these protein interactions are key for genome maintenance.

Our proposed model in Fig. 7 posits that in the G1 phase, the ORC complex, comprising ORC1-6, is loaded onto the replication origin. Following this, the licensing factors, Cdc6 and Cdt1 chaperones are recruited onto the origin (Takara and Bell, 2011). Through an interaction involving Shu complex DNA binding partners Csm2 and Psy3, the Shu complex is recruited to the replication origin, interacting with the ORC complex in G1. Concurrently, there is sequential recruitment and head-to-head binding of two inactive MCM2–7 hexamers onto the DNA by Cdc6 and Cdt1 (Schmidt et al, 2022). As S phase commences, the activation of the MCM complex is initiated by CDK/DDK. Facilitated by the Shu complex interaction with MCM complex and Cdc45, we propose that the Shu complex may accompany the replication fork and the CMG helicase (comprising Cdc45, MCM complex, and GINS) during replication initiation. This partnership facilitates the bypass of replication fork-blocking lesions, such as abasic sites. Although future experiments are needed to definitively demonstrate that the Shu complex travels with the replisome.

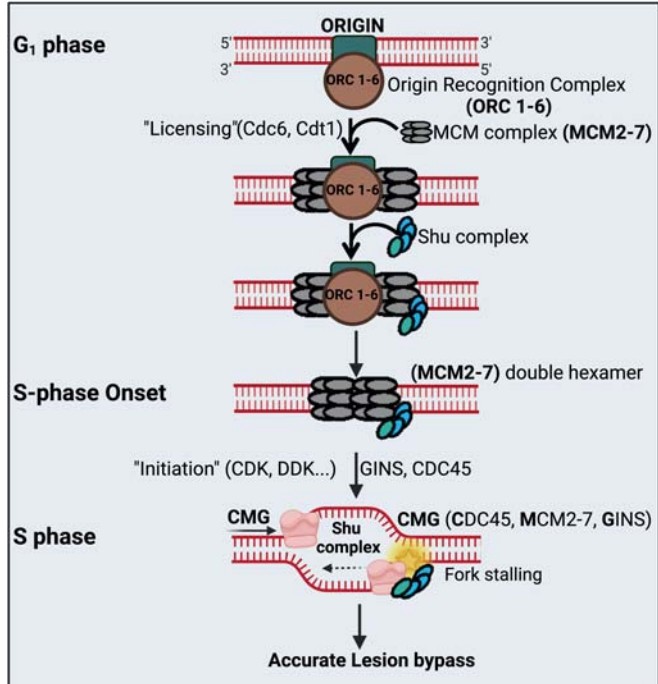

**Figure 7. Model of Shu complex function to bypass replicative damage by interacting with the replication machinery.**

During G1, the Origin Recognition Complex (ORC), consisting of ORC1-6 (green and brown ovals), is loaded onto the replication origin. Subsequently, the "licensing" factors Cdc6 and Cdt1 are recruited. The MCM complex, consisting of MCM2-7 (gray ovals) is recruited and a second MCM complex is loaded by the licensing factors Cdc6 and Cdt1. The Shu complex (blue and green ovals) is recruited to the replication origin through an interaction between the Shu complex member, Psy3, and the ORC complex. Subsequently, The Shu complex interacts with the MCM complex through Psy3 and recruits the other HR machinery, Rad51 and Rad52. During the S phase, the MCM complex is activated by the CDK/DDK, and the Shu complex interacts with the replication fork and the CMG helicase (consisting of Cdc45, MCM complex, and GINS), via Mcm4 to enable bypass of the replication fork blocking lesions (i.e., abasic sites). The Shu complex interaction with the replication machinery enables a Rad51-mediated template switch and subsequent gap repair. The figure is created with www.biorender.com.

Alternatively, the Shu complex may facilitate early S phase initiation at the origin to enable lesion bypass. Notably, in S phase, the Shu complex engages with the MCM complex via Psy3 in a DNA damage-independent manner, and further recruits additional HR machinery, Rad51, and Rad52. In support of this model, Mcm4 physically interacts with both Rad51 and Rad52 and these interactions are stabilized by Cdc7 (Cabello-Lobato et al, 2021; Cano-Linares et al, 2021). Studies in *Schizosaccharomyces pombe* and human cells have also identified a physical interaction between Rad51 and Rad52 proteins with MCM4, in a DNA damage-independent manner (Bailis et al, 2008; Shukla et al, 2005). Therefore, it appears they possess a conserved function. More recently, another non-canonical role of the replicative helicase was established where the MCM complex was shown to physically interact with the ribonucleotide reductase complexes (RNR) to enhance the dynamics of Rad52 during DNA damage (Yáñez-Vilches et al, 2024). Here, we propose that the Shu complex's interaction with the replication machinery plays a crucial role in facilitating a Rad51-mediated template switch, to facilitate error-free bypass of the DNA damage (Godin et al, 2016b, 2013; Shor et al, 2005). Importantly, these physical interactions with the replication initiation complexes occur independently of other HR machinery, including the recombinase Rad51 and the canonical Rad51 paralog, Rad55. Intriguingly, Csm2 enrichment at ARS sites is largely dependent on its interaction with Rad55. Interestingly, Rad55 is needed for Csm2 enrichment at ARS sites while being dispensable for Shu complex interaction with Mcm4. These results are consistent with those from the Prado laboratory showing that Mcm4 interaction with Rad51 or Rad52 is also DNA-independent (Cabello-Lobato et al, 2021). Furthermore, we show that Csm2 and Psy3 DNA binding is largely dispensable for its interaction with members of the MCM or ORC complexes (Fig. 3I,J). Therefore, it is possible that Rad55 helps to stabilize or enrich the Shu complex to ARS sites but that the Shu complex alone is needed to interact with the replisome. Overall, our results delineate a model wherein the Shu complex interacts with the replication machinery to ensure an error-free bypass of DNA damage. In the future, it will be important to determine if the Shu complex influences replication origin activation or replisome assembly and to delineate how the Shu complex moves or travels with the replication machinery to enable DNA lesion bypass.

High-fidelity DNA replication is essential for maintaining genomic stability (Cheng et al, 2012). However, the replication initiation process creates mutations at replication origins, which shapes the evolution of origin sequences and molds the human genome's regulatory domains (Murat et al, 2022). Notably, the processes causing mutagenesis during replication have been associated with ongoing replication forks or conflicts between transcription and replication (Jinks-Robertson and Bhagwat, 2014). These events lead to DNA damage and mutations distributed across distinct domains within the human genome. A recent examination of the distribution of germline mutations in humans has revealed distinct mutational processes linked to ongoing replication. These processes arise from the specific resolution of bulky DNA damage, such as those repaired by the Shu complex, and replication errors on specific DNA strands (Murat et al, 2022). Future studies should examine if loss of the human Shu complex impacts mutagenesis at replication origins in humans.

Consistent with a role in bypassing DNA damage during replication, Martino et al identified a restart defect in human Shu complex members, SWS1 and SWSAP1 CRISPR/Cas9 knockout RPE-1 cells upon replication fork stalling (Martino et al, 2019). Stalled replication forks can form ssDNA gaps resulting in genomic rearrangements including deletions and duplications (Larsen et al, 2017). This type of copy number variation occurs through HR repair following replication fork stalling and is observed in human cancers. Further links between replication and HR have been uncovered where the knockdown of a paralog of human MCM complex, MCM8 or MCM9, substantially decreases the efficiency of HR repair (Park et al, 2013). Suggesting that the MCM8-9 complex also plays an important role in DNA repair, MCM8-9 enables HR-mediated restart of stalled replication forks and is also activated by DSB induction (Natsume et al, 2017; McKinzey et al, 2021). Furthermore, RAD51 plays an essential role during replication by enabling the helicase to maintain a poised state, ready to resume unwinding and support DNA synthesis once the replication stress is resolved (Liu et al, 2023). It will be important to determine if the human N-terminus of the Shu complex also interacts with the replication initiation machinery acting as an anchor point with the replicative helicase and whether this has an analogous function in lesion bypass upon replication stress.

# Methods

**Reagents and tools table**

| Strains and plasmids used in this study | | | |
|---|---|---|---|
| **Name** | **Strain Number** | **Genotype** | **Source** |
| WT | W9100-12C | MATα ADE2 his3-11,15 leu2-3,112 LYS2 trp1-1 ura3-1 RAD5 | (Godin et al, 2016b) |
| WT | W9100-17D | MATa ADE2 his3-11,15 leu2-3,112 lys2Δ trp1 ura3-1 RAD5 | (Godin et al, 2015) |
| WT | KBY 138 | MATa ADE2/ade2-1 leu2-3,112/leu2-3,112 his3-11,15/his3-11,15 ura3-1/ura3-1 TRP1/trp1-1 LYS2/lys2Δ RAD5/RAD5 | This study |
| Csm2-6HA | KBY809-4C | MATa Csm2-6HA-k.l.TRP1 ADE2 LYS2 | (Godin et al, 2016b) |
| Psy3-6HA | KBY565-1A | MATa PSY3-6HA-KITRP1 ADE2 his3-11,15 LYS2 leu2-3,112 trp1-1 ura3-1 RAD5 | (Rosenbaum et al, 2019) |
| Shu1-6HA | KBY1077-1 | MATα shu1-6HA-k.l.TRP1 ADE2 his3-11,15 leu2-3,112 LYS2 ura3-1 trp1-1 RAD5 | This study |
| Shu1-6HA psy3Δ | KBY1097-2D | MATa shu1-6HA-K.l.TRP1 psy3::KanMX4 his3-11,15 ADE2 leu2-3,112 LYS2 ura3-1 RAD5 | This study |
| Shu2-6HA | KBY563-1A | MATa Shu2-6HA-K.l.TRP1 ADE2 his3-11,15 leu2-3,112 LYS2 trp1-1 ura3-1 RAD5 | This study |
| Shu2-6HA bar1Δ | KBY691-23C | MATa Shu2-6HA-K.l.TRP1 bar1::LEU2 ADE2 | This study |
| Csm2-KRRR-6HA | KBY1106-1 | MATa csm2-K189A,R190A,R191A,R192A-6HA-K.l. TRP1 LYS2 trp1-1 | (Rosenbaum et al, 2019) |
| Csm2-6HA rad51Δ | ADM 47 | MATa Csm2-6HA-k.l.TRP1 rad51::HIS3 ADE2 LYS2 | This study |

## Strains and plasmids used in this study

| Name | Strain Number | Genotype | Source |
|---|---|---|---|
| *Csm2-6HA rad52Δ* | ADM 57 | MAT**a** Csm2-6HA-k.l.TRP1 rad52::HIS3 ADE2 LYS2 | This study |
| *Csm2-6HA psy3Δ* | KBY871-1A | MAT**a** Csm2-6HA-k.l.TRP1 bar1::LEU2 psy3::KanMX4 ADE2 LYS2 | This study |
| *Csm2-F46A* | KBY611-1A | MATα csm2-F46A LYS2 trp1-1 | (Gaines et al, 2015) |
| *Csm2-6HA apn1Δ apn2Δ ntg1Δ ntg2Δ* | KBY1179-17D | MAT**a** Csm2-6HA-k.l.TRP1 apn1::hphNT1 apn2::hphNT1 ntg1::NatNT2 ntg2::NatNT2 ADE2 LYS2 trp1-1 | (Rosenbaum et al, 2019) |
| *Rad55-9Myc* | KBY1118-9C | MAT**a** Rad55-9MYC-hphMX4 ADE2 LYS2 trp1-1 | This study |
| *Rad55-9Myc Csm2-F46A* | KBY1111-1D | MAT**a** csm2-F46A-6HA-k.l.TRP1 rad55-9MYC-hphMX4 ADE2 LYS2 | This study |
| *Csm2-6HA psy3Δ* | KBY1096-3A | MAT**a** Csm2-6HA-k.l.TRP1 psy3::KanMX4 ADE2 his3-11,15 leu2-3,112 LYS2 ura3-1 RAD5 | This study |
| Y2H strain | PJ69-4α | MATα LYS2::GAL1-HIS3 GAL2-ADE2 met2::GAL7-lacZ gal4Δ gal80Δ his3-200 leu2-3,112 trp1-901 ura3-52 | (James et al, 1996) |
| *rad51Δ* | KBY215 | MATα LYS2::GAL1-HIS3 GAL2-ADE2 met2::GAL7-lacZ rad51::HPH4 gal4Δ gal80Δ his3-200 leu2-3,112 trp1-901 ura3-52 | This study |
| *rad55Δ* | KBY221 | MATα LYS2::GAL1-HIS3 GAL2-ADE2 met2::GAL7-lacZ rad55::HPH4 gal4Δ gal80Δ his3-200 leu2-3,112 trp1-901 ura3-52 | (Gaines et al, 2015) |
| *csm2Δ* | KBY395 | MATα LYS2::GAL1-HIS3 GAL2-ADE2 met2::GAL7-lacZ csm2::HPHT1 gal4Δ gal80Δ his3-200 leu2-3,112 trp1-901 ura3-52 | (Gaines et al, 2015) |
| *psy3Δ* | KBY220 | MATα LYS2::GAL1-HIS3 GAL2-ADE2 met2::GAL7-lacZ psy3::NAT1 gal4Δ gal80Δ his3-200 leu2-3,112 trp1-901 ura3-52 | This study |
| *shu1Δ* | KBY403 | MAT**a** LYS2::GAL1-HIS3 GAL2-ADE2 met2::GAL7-lacZ shu1::NatMX gal4Δ gal80Δ his3-200 leu2-3,112 trp1-901 ura3-52 | This study |
| *shu2Δ* | KBY 223 | MAT**a** LYS2::GAL1-HIS3 GAL2-ADE2 met2::GAL7-lacZ shu2::HYG gal4Δ gal80Δ his3-200 leu2-3,112 trp1-901 ura3-52 | This study |
| *psy3Δ for DRR* | KBY57-1B | MaT**a** leu2ΔEcoRI::URA3-HO::leu2ΔBstEII psy3::KanMX4 ade2-1 his3-11 LYS2 trp1-1 RAD5 | (Godin et al, 2015) |
| *psy3Δ CAN1* | KBY758-11A | MAT**a** psy3::KanMX4 ADE2 CAN1 LYS2 trp1-1 | (Rosenbaum et al, 2019) |

## Plasmids

| Name | Purpose | Backbone | Selection marker | Source |
|---|---|---|---|---|
| pGBD Psy3 | Y2H | pGBD | Kanamycin/ TRP | (Godin et al, 2013) |

## Plasmids

| Name | Purpose | Backbone | Selection marker | Source |
|---|---|---|---|---|
| pGBD Csm2 | Y2H | pGBD | Kanamycin/ TRP | (Godin et al, 2013) |
| pGBD Shu1 | Y2H | pGBD | Ampicillin/ TRP | This study |
| pGBD Shu2 | Y2H | pGBD | Ampicillin/ TRP | This study |
| pGAD Psy3 | Y2H | pGAD | Ampicillin/ LEU | (Gaines et al, 2015) |
| pGAD Csm2 | Y2H | pGAD | Ampicillin/ LEU | (Gaines et al, 2015) |
| pGAD Shu1 | Y2H | pGAD | Ampicillin/ LEU | This study |
| pGAD Shu2 | Y2H | pGAD | Ampicillin/ LEU | (Gaines et al, 2015) |
| pGBD Rad55 | Y2H | pGBD | Ampicillin/ TRP | (Gaines et al, 2015) |
| pGBD Rad57 | Y2H | pGBD | Ampicillin/ TRP | This study |
| pGAD Rad55 | Y2H | pGAD | Ampicillin/ LEU | (Godin et al, 2013) |
| pGAD Rad57 | Y2H | pGAD | Ampicillin/ LEU | (Godin et al, 2013) |
| pGAD Mcm2 | Y2H | pGAD | Ampicillin/ LEU | This study |
| pGAD Mcm3 | Y2H | pGAD | Ampicillin/ LEU | This study |
| pGAD Mcm4 | Y2H | pGAD | Ampicillin/ LEU | This study |
| pGBD Mcm4 | Y2H | pGBD | Ampicillin/ TRP | This study |
| pGAD Mcm5 | Y2H | pGAD | Ampicillin/ LEU | This study |
| pGAD Mcm6 | Y2H | pGAD | Ampicillin/ LEU | This study |
| pGAD Mcm7 | Y2H | pGAD | Ampicillin/ LEU | This study |
| pGBD Mcm7 | Y2H | pGBD | Ampicillin/ TRP | This study |
| pGBD Mcm10 | Y2H | pGBD | Ampicillin/ TRP | This study |
| pGBD csm2-K189A, R190A, R191A, R192A | Y2H | pGBD | Ampicillin/ TRP | This study |
| pGBD psy3-K199A, R200A, K201A | Y2H | pGBD | Ampicillin/ TRP | This study |
| pGAD ORC1 | Y2H | pGAD | Ampicillin/ LEU | This study |
| pGAD ORC2 | Y2H | pGAD | Ampicillin/ LEU | This study |

| Plasmids | | | | |
| --- | --- | --- | --- | --- |
| Name | Purpose | Backbone | Selection marker | Source |
| pGAD ORC3 | Y2H | pGAD | Ampicillin/ LEU | This study |
| pGAD ORC4 | Y2H | pGAD | Ampicillin/ LEU | This study |
| pGAD ORC5 | Y2H | pGAD | Ampicillin/ LEU | This study |
| pGAD ORC6 | Y2H | pGAD | Ampicillin/ LEU | This study |
| pGAD CDC45 | Y2H | pGAD | Ampicillin/ LEU | This study |
| pGBD psy3-Y10A | Y2H | pGBD | Ampicillin/ TRP | This study |
| pGBD psy3-L12A | Y2H | pGBD | Ampicillin/ TRP | This study |
| pGBD psy3-F15A | Y2H | pGBD | Ampicillin/ TRP | This study |
| pGBD psy3-T17A | Y2H | pGBD | Ampicillin/ TRP | This study |
| pGBD psy3-S18A | Y2H | pGBD | Ampicillin/ TRP | This study |
| pRS413 psy3-Y10A | | pRS413 | Ampicillin/ HIS | This study |
| pRS413 psy3-L12A | | pRS413 | Ampicillin/ HIS | This study |
| pRS413 psy3-F15A | | pRS413 | Ampicillin/ HIS | This study |
| pRS413 psy3-T17A | | pRS413 | Ampicillin/ HIS | This study |
| pRS413 psy3-S18A | | pRS413 | Ampicillin/ HIS | This study |
| pRS413 psy3-WT | | pRS413 | Ampicillin/ HIS | This study |
| pGAD-C1 | Y2H empty vector | N/A | LEU | (Godin et al, 2013) |
| pGBD-C1 | Y2H empty vector | N/A | TRP | (Godin et al, 2013) |
| Yiplac211-psy3-Y10A | Yeast integration | Yiplac211 | Ampicillin/ URA3 | This study |

| PCR Oligos for ChIP | | | |
| --- | --- | --- | --- |
| Name | Sequence 5′-3′ | | Purpose |
| ARS216 Fw | TCCGCGCTAGAATCTGGAAT | | Positive locus |
| ARS216 Rv | CCTCTTCTTCGCTTCTTCGC | | Positive locus |
| ARS306 FW | AGCCGACCTATCCTATGC | | Positive locus |
| ARS306 Rv | CTCCTTAGTAGTCCACAGTTC | | Positive locus |
| ARS305 Fw | CAGTATTTCAGGCCGCTCTT | | Positive locus |
| ARS305 Rv | GTCTTGCTGTGGCCTCAATC | | Positive locus |
| ChrIV IG Fw | CGAAGTATACCGTGCGTC | | Gene desert on ChrIV |
| ChrIV IG Rv | AGCTTCTTGCTGCTCTATG | | Gene desert on ChrIV |

| Chemical peptides and recombinant proteins | | |
| --- | --- | --- |
| Name | Source | Identifier |
| Benzonase | Millipore Sigma | E1014-5KU |
| DNase I | BioLabs | M0303S |
| Pepsin | Sigma | P6887-1G |
| α-factor | GenScript | RP01002 |
| Nocodazole | Sigma | M1404-10MG |
| Anti-HA Magnetic Beads | Thermo Scientific | 88837 |
| PhosSTOP (Roche) | Millipore Sigma | 4906837001 |
| cOmplete Mini, EDTA-free | Millipore Sigma | 11836170001 |
| Pierce Classic Magnetic IP/Co-IP Kit | Thermo Scientific | 88804 |
| Methyl-methanesulfonate (MMS) | Sigma-Aldrich | 129925-25G |
| Propidium iodide solution | Sigma-Aldrich | P4864-10MLS |
| Pronase | Roche | 10165921001 |
| L-Canavanine sulfate salt | Sigma-Aldrich | C9758-1G |
| NEBNext® Ultra™ II DNA Library Prep Kit for Illumina® | New England Biolabs | NEB ref. E7645L |
| Pepsin from porcine gastric mucosa | Sigma-Aldrich | P6887-1G |
| 5-Fluoroorotic Acid | Thermo Scientific | 2837984 |
| Dynabeads Protein G beads | Thermo Scientific | 10009D |
| **Antibodies** | | |
| Mouse monoclonal anti-Orc2 | Santa Cruz | Cat# sc-398410 |
| Mouse monoclonal anti-Mcm4 | Santa Cruz | Cat# sc-166036 |
| Rabbit polyclonal anti-HA | Abcam | Cat# ab9110 |
| Rabbit polyclonal anti-Clb2 | Santa Cruz | Cat# sc-9071 |
| Rabbit polyclonal anti-Kar2 | Santa Cruz | Cat# sc-33630 |
| Rabbit polyclonal anti-GAPDH | Santa Cruz | Cat# sc-25778 |
| Mouse monoclonal anti-Myc | Takara | |

The strains are listed in the order they appear in the figures and text. The yeast strain used in all experiments was W303-1A except for the Y2H experiments which were performed in either PJ69-4A or PJ69-4α. All yeast growth was performed at 30 °C; all bacterial growth was performed at 37 °C. Bacterial transformants were selected on a standard LB medium with ampicillin (100 µg/mL). Yeast transformants were selected on standard synthetic complete medium lacking uracil.

## Yeast strains, plasmids, oligonucleotides

All yeast strains are derived from W303 (W1588-4C and W5059-1B) and are *RAD5+* from (Thomas and Rothstein, 1989) and W5059-1B (Zhao et al, 1998). Yeast-two-hybrid assays were performed in PJ69-4α as described by (Godin et al, 2013) and (James et al, 1996). All strains and plasmids used in this study can be found in the "Reagents and Tools" Table. A list of primers used for ChIP experiments is listed in the "Reagents and Tools" Table. The MCM complex members

(MCM2-7) and MCM10 are cloned in both the GAL4 DNA binding (pGBD-C1) domain and GAL4 DNA activating (pGAD-C1) domain vectors using the *EcoRI* (GAATTC)-*BamHI* (GGATCC) cloning sites by Gene Universal. The ORC complex members (ORC1-6) and CDC45 were cloned in pGAD-C1 using the *EcoRI* (GAATTC)-*BamHI* (GGATCC) cloning sites by Gene Universal. Site-directed mutagenesis was used to create mutations in the Psy3 N-terminal region. The identified Psy3 mutants (*psy3-Y10A, L12A, F15A, T17A, S18A*) were PCR cloned in the pGBD-C1 vector at the *EcoRI, BamHI* cloning sites. The *psy3-Y10A* is also integrated into the Yiplac211 vector at the *HindIII, EcoRI* cloning site.

## Myc/HA-chromatin Immunoprecipitation

Myc/HA-ChIP assays were performed using a protocol adapted from previously described methods (De Magis et al, 2020). In summary, cells were lysed with the aid of glass beads in a Fastprep-24 homogenizer, and chromatin was fragmented to sizes ranging between 200 and 1000 base pairs using a Bioruptor® Pico sonicator (Diagenode) under high-intensity settings (30 s ON, 30 s OFF, for 7 cycles). The efficiency of chromatin shearing was evaluated by running samples on a 1% agarose gel. To immunoprecipitate target proteins, 8 µg of anti-Myc antibody (Takara) or 3 µg of anti-HA antibody (Santa Cruz, catalog #sc-7392) was added to the chromatin and incubated at 4 °C for 2 h. Following this, the antibody-bound complexes were captured by incubating with 80 µl Dynabeads-Protein G (Thermo Scientific) for an additional 2 h at 4 °C. Beads were subsequently washed three times using a washing buffer containing 100 mM KCl, 0.1% (w/v) Tween-20, and 1 mM Tris-HCl (pH 7.5). DNA was eluted from the beads and purified using the MinElute Kit (Qiagen). The purified DNA was then analyzed by quantitative PCR (qPCR) using primers listed in the "Reagents and Tools" table.

## ChIP-seq analysis

Chromatin immunoprecipitation followed by sequencing (ChIP-seq) was performed as outlined in the methods above. DNA libraries for genome-wide sequencing were prepared using the NEBNext Ultra II DNA Library Prep Kit for Illumina (NEB) following the manufacturer's guidelines. The sequencing was carried out on a HiSeq 2500 platform. Sequence reads were aligned to the *saccharomyces cerevisiae* reference genome (sacCer3) using Bowtie (Langmead et al, 2009). Binding regions were identified with MACS 2.0 software, optimized for narrow peak detection, using default parameters (Zhang et al, 2008). ChIP input samples were included as control datasets. A custom PERL script incorporating permutation analysis was used to evaluate overlaps between identified binding sites and other genomic features, providing insights into potential functional associations.

## Statistical analysis

All statistical comparisons were performed using either a one-sided Student's t-test, one-way ANOVA, or Tukey test. Statistical significance relative to wild-type cells was as follows: *$p < 0.05$, **$p < 0.01$, ***$p < 0.001$, and ****$p < 0.0001$ except stated otherwise in figure legend. The data presented represents the mean values from at least three independent biological experiments ($n = 3$ or more).

## Evolutionary rate covariation analysis

Evolutionary rate covariation was calculated for each member of the Shu complex between all members of the MCM, ORC, GINS, MTC, and CMG complexes. The analysis was performed for a dataset of 343 yeast species (Shen et al, 2018) and 63 mammalian species. The ERC value for each protein pair is calculated by performing a Pearson correlation on the relative evolutionary rates (Partha et al, 2017) of all shared branches for that gene pair. Before performing the correlation, the relative evolutionary rates are Winsorized, condensing the 3 most extreme values to the 4th, removing potential outliers that drive high correlation. After the Pearson correlation, the correlation value is Fisher transformed which normalizes the values based on the number of branches that contribute to the score. This also changes the potential spread from $[-1,1]$ to $[-\text{infinity}, \text{infinity}]$. Permutation tests were performed to compare the ERC values between the Shu complex and all other complexes to a genome-wide null distribution. For the yeast dataset, this comprises creating a null distribution from 1000 randomly selected subsets of genes equal in size to the Shu vs complex dataset from a genome-wide matrix of 12,552 orthologous genes. Likewise, the mammalian Shu vs complex permutation test was performed on a genome-wide matrix of 19,149 orthologous genes, selecting 1000 random subsets as the null distribution. The code to run ERC can be found at: https://github.com/nclark-lab/erc

## Yeast-two-hybrid assays

Yeast two-hybrid experiments were conducted by co-transforming the designated pGAD and pGBD/pGBK plasmids into the yeast strain *PJ69-4α*, following a previously established protocol by (Godin et al, 2013). For the transformations, yeast cultures were grown in YPD medium to mid-log phase (OD600 between 0.3 and 0.6), collected by centrifugation, and resuspended in a solution containing 100 mM lithium acetate and 1× Tris-EDTA along with 300 ng of plasmid DNA. Transformation mixtures were incubated at 30 °C for 30 min with 34% polyethylene glycol (PEG), followed by a 15-min heat shock at 42 °C. Cells were then plated onto a selective synthetic complete medium lacking leucine and tryptophan (SC-L-W) to select for transformants carrying the plasmids, which contained leucine and tryptophan auxotrophic markers, respectively. Three independent transformants were grown in 5 mL SC-L-W medium at 30 °C overnight and diluted to an initial $OD_{600}$ of 0.2 in 3 mL SC-L-W medium. After incubation for 4 h, cultures were adjusted to an equivalent cell density ($OD_{600} = 0.5$), and five µL of each culture were spotted onto SC-L-W plates (as a loading control) or SC-L-W-H plates (to test for yeast two-hybrid interactions). Plates were incubated at 30 °C for 2–3 days, and images were captured after incubation. Brightness and contrast adjustments of the images were performed using Adobe Photoshop. Each experiment was independently replicated 3–4 times.

## Western blot analysis

Cells expressing Csm2-6HA, Psy3-6HA, Shu1-6HA, or Shu2-6HA were cultured overnight at 30 °C in 5 mL YPD media. The cultures were diluted to an OD600 of 0.2 in 100 mL of fresh YPD media and incubated at 30 °C for 2 h. To obtain an asynchronous control

sample, cells equivalent to 0.75 OD600 were collected and set aside for further analysis. For G1 arrest, the cultures were treated with 2 mg/mL α-factor and incubated for 2 h at 30 °C. Successful G1 arrest was confirmed by examining cell morphology (shmoo formation) and performing flow cytometry, as described below. Following arrest, the cells were washed three times with cold water and resuspended in 50 mL of fresh YPD media. Samples (0.75 OD$_{600}$) were taken every 20 min over a 120-min time course, pelleted, washed with water, and flash-frozen in a dry ice/ethanol bath. For flow cytometry, an additional sample of 0.75 OD$_{600}$ was resuspended in 70% ethanol after pelleting and washing.

Whole-cell lysates were prepared from equal numbers of cells (0.75 OD600) using TCA precipitation. The resulting protein extracts were resuspended in 50 μL of sample buffer containing HU, DTT, and Tris, as previously described (Koontz, 2014; Foiani et al, 1994). Protein samples (10 μL) were separated on a 10% SDS-PAGE gel at 100 V for 2 h and transferred to a nitrocellulose membrane using a semidry transfer apparatus at 15 V for 3 h (BioRad). Detection of specific proteins was carried out using the following primary antibodies: HA (1:2500, Abcam ab9110) to detect Psy3, Csm2, Shu1, and Shu2; Clb2 (1:500, Santa Cruz sc-9071) as a marker for G2/M phase; Kar2 (1:200, Santa Cruz sc-33630); and GAPDH (1:10,000, Santa Cruz sc-25778) as a loading control. Secondary antibodies were used at a dilution of 1:10,000. Blots were visualized using standard imaging methods, and the resulting films were processed using Photoshop (Adobe Systems Incorporated) for adjustments to contrast and brightness. Source data files include unprocessed and uncropped images of the films.

## Flow cytometry and budding analysis

To assess DNA content, flow cytometry was conducted on cells fixed with 70% ethanol. Fixed cells were washed with phosphate-buffered saline (PBS), treated with RNase A (1 mg/ml in PBS) to degrade RNA, and stained with propidium iodide (5 mg/ml) to visualize DNA. Before analysis, samples were sonicated to ensure separation into single-cell suspensions and analyzed using a FACSCalibur flow cytometer. The proportion of budded cells was calculated by manually counting 100 cells per time point across independent experiments.

## Co-immunoprecipitation

Yeast cells were cultured at 30 °C in yeast extract peptone dextrose (YPD) medium. For synchronization experiments in the G2/M phase, cells were treated with 15 mg/mL nocodazole (Sigma-Aldrich) for 1 h in a YPD-rich medium. G1 synchronization was achieved by growing cells to mid-log phase, followed by the addition of 2 mg/mL α-factor (Genscript) for 2 h. Synchronization in G1 was confirmed by observing shmoo formation and performing flow cytometry analysis as described above. After arrest in G1, cells were washed three times and resuspended in 50–100 mL of fresh YPD medium, with or without 0.03% methyl methane-sulfonate (MMS), and incubated for 45 min.

Co-immunoprecipitation (CoIP) was conducted using 100 mL of cultures at mid-log phase (~0.5 OD600). Cells were lysed using ~200 μL of glass beads in 0.5 mL NP-40 lysis buffer (50 mM Tris-HCl pH 7.5, 100 mM NaCl, 2 mM MgCl2, 10% glycerol, and 0.1% NP-40) supplemented with protease inhibitors (1 mM PMSF,

Roche Complete EDTA-free) and phosphatase inhibitors (Roche PhosSTOP). The lysis process was carried out at 4 °C for 5 min. Cell debris was removed by centrifugation at $5000 \times g$ for 5 min at 4 °C. For experiments requiring nuclease treatment, the lysate was incubated with either 15 U of DNase I or 25 U of Benzonase at 37 °C for 20 min before further processing.

The supernatant was collected after centrifugation at $20,000 \times g$ for 25 min at 4 °C, and the protein concentration was measured using a Bradford assay. An aliquot (25 μL) from each sample was set aside as input. For immunoprecipitation of Mcm4 and Orc2-HA, the lysate was incubated overnight at 4 °C with 25 μL of Pierce Anti-HA Magnetic Beads. The beads were washed thoroughly using a modified NP-40 lysis buffer (50 mM Tris-HCl pH 7.5, 1 M NaCl, 1% NP-40). Proteins bound to the beads were analyzed by western blot using appropriate antibodies. For CoIP experiments involving MMS or nuclease-treated lysates, the same protocol was followed with additional steps as indicated.

## Serial dilution

Cells with a disrupted *PSY3* gene were transformed with either a plasmid carrying the wild-type PSY3 gene or a plasmid encoding the specified *PSY3* mutants. Transformants were selected on SC-TRP medium. Two independent transformants were grown overnight at 30 °C in SC-TRP medium until they reached the logarithmic growth phase. Cell cultures were adjusted to an optical density of 0.5 at 600 nm (OD$_{600}$), and equal numbers of cells were subjected to five-fold serial dilutions. Dilutions were then spotted onto a rich medium (YPD) without MMS or supplemented with MMS at concentrations of 0.01%, 0.02%, or 0.03%. Plates were incubated at 30 °C for 48 h, after which images of the colonies were captured.

## Canavanine mutagenesis assay

The protocol for the mutagenesis assay was adapted from methods described by (Godin et al, 2016b). Four independent *CAN1* colonies from *psy3Δ* cells transformed with pRS413 psy3-WT, pRS413 *psy3-Y10A*, or an empty vector were cultured overnight at 30 °C in 3 mL of SC-HIS medium, with or without 0.00033% MMS. After approximately 20 h of growth, cultures were diluted to an optical density at 600 nm (OD$_{600}$) of 3. A 150 μL aliquot of the undiluted culture was plated onto SC-ARG + CAN medium, and a 1:10,000 dilution was plated in 150 μL aliquots onto SC plates as a control. Plates were incubated at 30 °C for 48 h, after which colony growth was scanned and quantified using OpenCFU software (Geissmann, 2013). Mutation rates were determined by calculating the ratio of colonies growing on SC-ARG + CAN plates to the total colonies growing on SC plates. Median mutation frequencies were calculated from three independent experiments, and results are presented as the average frequency with standard deviations. Statistical significance was determined using a two-way ANOVA followed by Tukey's multiple comparison test.

## Direct repeat recombination assay

The mitotic recombination assay was conducted following the protocol described by (Godin et al, 2013), with modifications as necessary for this study. This assay involves two disrupted *leu2* heteroalleles, one cleaved by *EcoRI* and the other by *BsteII*, separated by an intervening *URA3* gene. Recombination events are measured by restoring a

functional *LEU2* gene, enabling yeast to grow on media lacking leucine. Briefly, nine independent colonies of each yeast strain were grown in SC-HIS medium (with or without 0.0003% MMS) at 30 °C for ~18 h. Cultures were adjusted to an optical density of 0.5 at 600 nm ($OD_{600}$), and 250 μl of this normalized culture was plated on SC-LEU plates to assess total recombination events. To determine cell viability, cultures were diluted 1:1000, and 250 μl of the diluted suspension was plated on SC plates. After incubating the plates at 30 °C for two days, colonies were counted. Subsequently, the SC-LEU plates were replica-plated onto SC-URA and SC with 5′-FOA (450 mg/L) to distinguish Rad51-dependent recombination events (gene conversion) from Rad51-independent recombination events (single-strand annealing. Recombination rates and standard deviations were calculated using an online colony-forming unit (CFU) analysis tool (Geissmann, 2013). Each experiment was performed in triplicate, and the average recombination rates from three independent experiments were calculated and visualized. Statistical significance was determined using Tukey's test.

## Data availability

ChIP-seq data have been uploaded to the National Center for Biotechnology Information (NCBI) Sequencing Read Archive under the reference number PRJNA1112421 and URL: https://www.ncbi.nlm.nih.gov/sra/?term=PRJNA1112421.

The source data of this paper are collected in the following database record: biostudies:S-SCDT-10_1038-S44318-025-00365-9.

## Peer review information

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

## Acknowledgements

The data for this manuscript were generated using the FACS analysis Shared Resource Laboratory at the University of Pennsylvania and is partially supported by the Abramson Cancer Center NCI Grant (P30 CA016520), Penn Center for Genome Integrity, and the Basser Center for BRCA Research, and the National Institute of Health (R01 ES030335) to KAB. Support for JHL and NLC was provided by an NHGRI grant (R01 HG009299). Research in the Paeschke laboratory is funded by the Deutsche Forschungsgemeinschaft (DFG, German Research Foundation) under Germany's Excellence Strategy – EXC2151 – 390873048 and CANTAR. The project "CANTAR" is receiving funding from the program "Netzwerke 2021", an initiative of the Ministry of Culture and Science of the State of Northrhine Westphalia. Further, this project is supported by a grant from the Fritz Thyssen Foundation to KP. ADM is supported by a postdoctoral fellowship funded by BONFOR (2020-1B-03, University Hospital Bonn). KG acknowledges the support of the Johnson Research Foundation.

## Author contributions

**Adeola A Fagunloye**: Conceptualization; Data curation; Formal analysis; Investigation; Methodology; Writing—original draft; Writing—review and editing. **Alessio De Magis**: Formal analysis; Investigation; Visualization; Writing—original draft. **Jordan H Little**: Data curation; Software. **Isabela Contreras**: Formal analysis; Methodology. **Tanis J Dorwart**: Formal analysis. **Braulio Bonilla**: Conceptualization. **Kushol Gupta**: Data curation; Software. **Nathan Clark**: Investigation; Methodology. **Theresa Zacheja**: Data curation. **Katrin Paeschke**: Conceptualization; Formal analysis; Funding acquisition; Investigation; Writing—original draft. **Kara A Bernstein**: Conceptualization; Formal analysis; Funding acquisition; Writing—original draft; Writing—review and editing.

Source data underlying figure panels in this paper may have individual authorship assigned. Where available, figure panel/source data authorship is listed in the following database record: biostudies:S-SCDT-10_1038-S44318-025-00365-9.

## Disclosure and competing interests statement

The authors declare no competing interests.

# Expanded View Figures

**Figure EV1.   Csm2, Rad55, and Rad52 significantly overlap at the same DNA binding regions by ChIP seq.**

(**A**) Five-fold serial dilution of the parental wild-type strain and strains expressing either Csm2-6HA or Rad55-9MYC on rich YPD medium or YPD medium containing the indicated concentration of MMS. (**B**) Genome-wide correlation of the Csm2 peaks with the RAD55 peaks. Exact *p*-value: $p = < 0.0001$. (**C**) Genome-wide correlation of the RAD55 peaks with the ARS sites. Exact *p*-value: $p = 0.36$. (**D**) Genome-wide correlation of the Csm2 peaks with the RAD52 peaks (Costantino and Koshland, 2018). Exact *p*-value: $p = < 0.0001$. (**E**) Genome-wide correlation of the RAD55 peaks with the RAD52 peaks (Costantino and Koshland, 2018). Exact *p*-value: $p = < 0.0001$. (**B–E**) Significance is assessed by genome-wide permutation test (**F**) IGV Genome Browser screenshot of genome-wide Csm2, RAD55, RAD52, (Costantino and Koshland, 2018), MCM2-7 binding sites at ChrII:611,265-613,215 (Lee et al, 2021). ARS database is from OriDB, produced in Saccer1. Source data are available online for this figure.

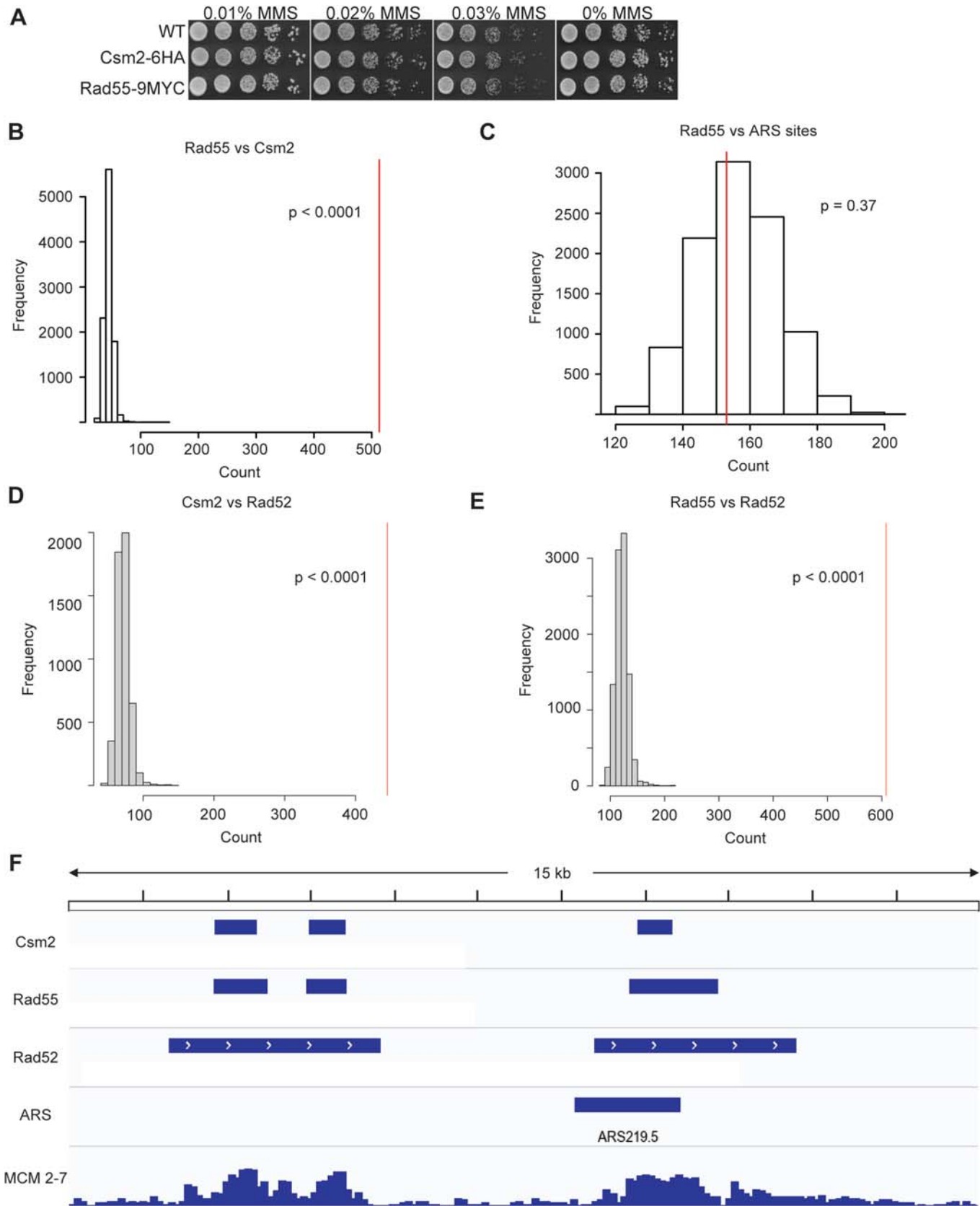

**A**

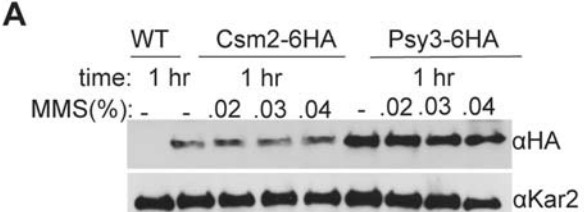

**B**

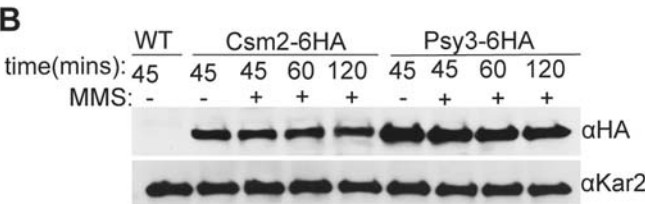

**Figure EV2. Csm2 and Psy3 steady-state protein levels are maintained upon MMS exposure.**

(A) Csm2 and Psy3 steady-state protein levels are not increased upon MMS exposure. Csm2-6HA or Psy3-6HA expressing strains were exposed to the indicated dose of MMS for one hour and then protein levels were accessed by western blot using αHA or αKar2 antibodies. Kar2 was used as a loading control. (B) Csm2 and Psy3 steady-state protein levels remain constant over time after 0.03% MMS exposure. Csm2-6HA or Psy3-6HA expressing strains were exposed to 0.03% MMS for the indicated amount of time (45, 60, or 120 min) and then protein levels were accessed by western blot using αHA or αKar2 antibodies. Kar2 was used as a loading control. Source data are available online for this figure.

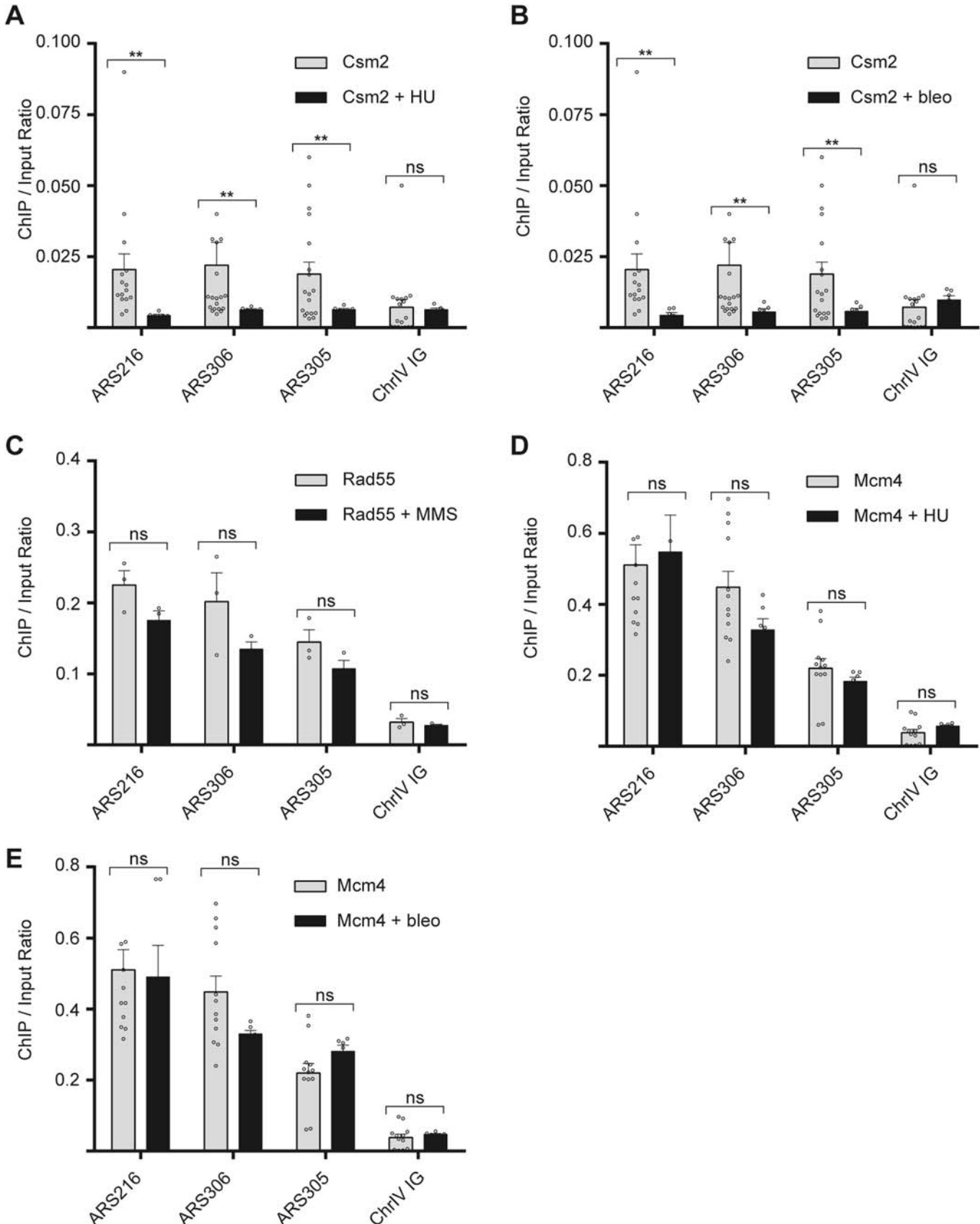

◀

**Figure EV3.  Unlike Rad55 or Mcm4, Csm2 enrichment at ARS sites is reduced upon DNA damage.**

(A) ChIP and qPCR of Csm2-6HA untreated or treated for 2 h with 100 mM HU. Exact *p*-value: ARS216 $p = 0.007797$, ARS305 $p = 0.001494$, ARS306 $p = 0.006935$, and ChrIV IG $p = 0.464314$. (B) ChIP and qPCR of Csm2-6HA untreated or treated for 2 h with 20 ng/μL bleomycin. Exact *p*-value: ARS216 $p = 0.002339$, ARS305 $p = 0.001465$, ARS306 $p = 0.003030$, ChrIV IG $p = 0.164196$. (C) ChIP and qPCR of RAD55-9MYC untreated or treated for 2 h with 0.02% MMS. Exact *p*-value: ARS216 $p = 0.163698$, ARS305 $p = 0.106164$, ARS306 $p = 0.051302$, and ChrIV IG $p = 0.307960$. (D) ChIP and qPCR of MCM4 untreated or treated for 2 h with 100 mM HU. Exact *p*-value: ARS216 $p = 0.386663$, ARS305 $p = 0.295602$, ARS306 $p = 0.089571$ and ChrIV IG $p = 0.327584$. (E) ChIP and qPCR of MCM4 untreated or treated for 2 h with 20 ng/μL bleomycin. Exact *p*-value: ARS216 $p = 0.280850$, ARS305 $p = 0.087874$, ARS306 $p = 0.188565$ and ChrIV IG $p = 0.446871$. For all experiments, qPCRs were performed at three ARS sites (ARS302, ARS306, ARS305) and one control (ChrIV IG). All plotted results were based on the ChIP/Input ratio average of at least three independent experiments ± SEM. Significance was calculated based on a one-sided Student's t-test. Asterisks indicate statistical significance in comparison with wild-type cells under the same experimental conditions. *$p < 0.05$, **$p < 0.01$ and ns = not statistically significant. Source data are available online for this figure.

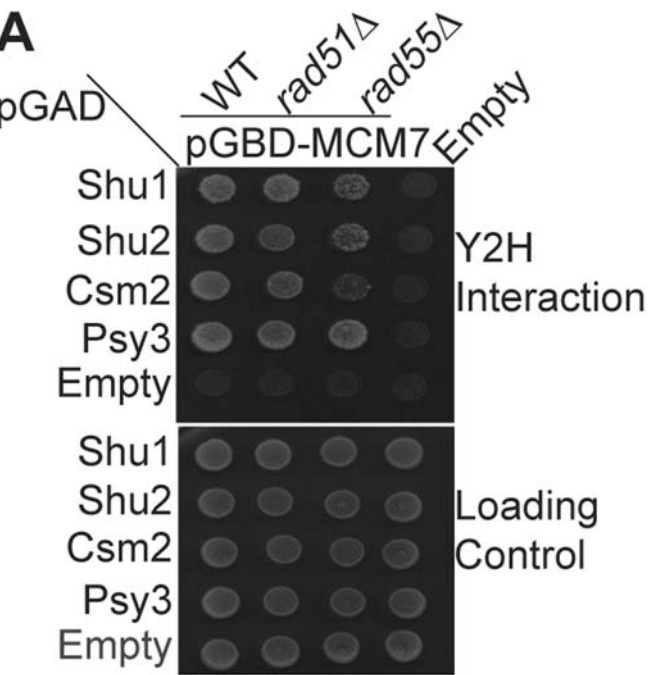

**Figure EV4.  Mcm7 Y2H interaction with the Shu complex is independent of RAD51 or RAD55.**

Shu complex Y2H interaction with MCM complex member (Mcm7) is independent of RAD51 and RAD55. Y2H experiment examining Shu complex (Shu1, Shu2, Csm2, Psy3) interaction with Mcm7 transformed in Y2H strain with either rad51Δ or rad55Δ knocked out. Yeast with the indicated plasmids were grown in SC-L-W, plated on SC-L-W-H medium, and incubated for 2–3 days at 30 °C. Empty vectors are negative controls. Growth is indicative of a Y2H interaction and SC-L-W is used as a loading control. All experiments were done in triplicate. Source data are available online for this figure.

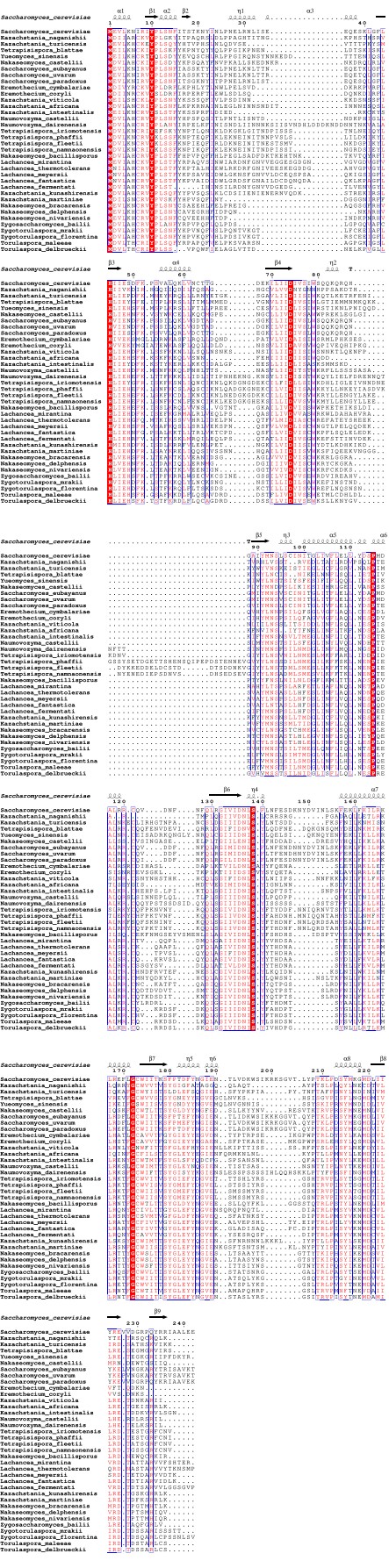

◀ **Figure EV5.  Psy3 sequence alignment from 36 fungi species.**

Sequence alignment of Shu complex member, Psy3, from the indicated 36 fungi species (Larkin et al, 2007). Invariant residues are shown in red boxes, and similar residues are shown in red text and outlined in blue. The predicted protein folding based on the *S. cerevisiae* structure (PDB: 5XYN) (Zhang et al, 2017) is shown above. The figure was made with ESPript3 (Robert and Gouet, 2014).

