## [Peer Review File · The EMBO Journal]

The Shu complex interacts with replicative helicase to prevent mutations and aberrant recombination

Adeola Fagunloye, Alessio De Magis, Jordan Little, Isabela Contreras, Tanis Dorwart, Braulio Bonilla, Kushol Gupta, Nathan Clark, Theresa Zacheja, Katrin Paeschke, and Kara Bernstein

Corresponding author(s): Kara Bernstein (kara.bernstein@penmedicine.upenn.edu) , Katrin Paeschke (katrin.paeschke@ukbonn.de)

Review Timeline:

Submission Date:	13th Mar 24
Editorial Decision:	8th Apr 24
Revision Received:	5th Nov 24
Editorial Decision:	3rd Dec 24
Revision Received:	18th Dec 24
Accepted:	8th Jan 25

Editor: Hartmut Vodermaier

Transaction Report:

Prof. Kara A Bernstein
University of Pennsylvania School of Medicine
Biochemistry and Biophysics
421 Curie Boulevard
BRB II/III Room 411
Philadelphia, PA 19104-6160

8th Apr 2024

Re: EMBOJ-2024-117270
RAD51 paralogs interact with the replicative helicase to bypass DNA damage

Dear Kara and Katrin,

Thank you again for submitting your work on Shu complex interactions with the replisome for our consideration. I sent it to three expert reviewers, who have now returned the reports copied below. All referees acknowledge the potential interest of your new findings, as well as the overall quality of the study, but they also raise several substantive concerns regarding physiological significance; the experimental support for certain key conclusions such as co-translocation of Shu and MCM complexes; and the depth of insight into several core aspects of the work.

Should you be able to satisfactorily address these key criticisms, as well as the more specific points listed in the reports, we would be interested in pursuing a revised manuscript further for EMBO Journal publication. Since our single-major-revision-round policy makes it important to diligently respond to each referee point at the time of resubmission, I would however encourage you to contact me with a preliminary point-by-point response already during the early stages of your revision work, in order to clarify how key issues may be addressed and to discuss possible revision plans (happily via Zoom if needed). We would also be open to extension of the default three-months revision period if needed; our 'scooping protection' (meaning that competing work appearing elsewhere in the meantime will not affect our considerations of your study) would of course remain valid also throughout such an extension.

Detailed information on preparing, formatting and uploading a revised manuscript can be found below and in our Guide to Authors. Thank you again for the opportunity to consider this work for The EMBO Journal, and I look forward to hearing from you in due time.

With kind regards,

Hartmut

3) Revised manuscript text (including main tables, and figure legends for main and EV figures) has to be submitted as editable

text file (e.g., .docx format). We encourage highlighting of changes (e.g., via text color) for the referees' reference.

4) Each main and each Expanded View (EV) figure should be uploaded as individual production-quality files (preferably in .eps, .tif, .jpg formats). For suggestions on figure preparation/layout, please refer to our Figure Preparation Guidelines:

8) Please note that supplementary information at EMBO Press has been superseded by the 'Expanded View' for inclusion of additional figures, tables, movies or datasets; with up to five EV Figures being typeset and directly accessible in the HTML version of the article. For details and guidance, please refer to:

embopress.org/page/journal/14602075/authorguide#expandedview

9) Digital image enhancement is acceptable practice, as long as it accurately represents the original data and conforms to community standards. If a figure has been subjected to significant electronic manipulation, this must be clearly noted in the figure legend and/or the 'Materials and Methods' section. The editors reserve the right to request original versions of figures and the original images that were used to assemble the figure. Finally, we generally encourage uploading of numerical as well as gel/blot image source data; for details see: embopress.org/page/journal/14602075/authorguide#sourcedata

At EMBO Press, we ask authors to provide source data for the main manuscript figures. Our source data coordinator will contact you to discuss which figure panels we would need source data for and will also provide you with helpful tips on how to upload and organize the files.

Further information is available in our Guide For Authors:

In the interest of ensuring the conceptual advance provided by the work, we recommend submitting a revision within 3 months (7th Jul 2024). Please discuss the revision progress ahead of this time with the editor if you require more time to complete the revisions. Use the link below to submit your revision:

Link Not Available

Referee #1:

In 'RAD51, paralogs interact with the replicative helicase to bypass DNA damage. The authors present evidence that members of the SHU complex can directly interact with the origin of the replication complex. The SHU complex is important to DNA repair and comprises several Rad51 paralogs and an additional swim domain protein. This complex is conserved across eukaryotes. In general, the SHU complex works with other Rad51 paralogs, Rad55/57, to promote DNA repair. It is important in dealing with replication blocks, and this study provides strong evidence that subunits of the SHU complex interact with the MCM complex at ARS sites in *S. cerevisiae*. The key finding from this manuscript is that the SHU complex is recruited to and potentially travels with the replicative helicase during DNA synthesis. The authors further characterize this interaction and identify a region on the SHU protein Psy3 that forms the critical interaction with MCM. This provides a novel hypothesis for how SHU may aid lesion bypass during DNA synthesis.

The findings in this paper are exciting, novel, and of general interest to a broad readership. The experiments are of high quality, and the data in the manuscript generally supports the model that the SHU complex interacts directly with the ORC and that this interaction is specific and necessary for function. To this reviewer, it is still unclear how well the evidence supports the idea that SHU complex members travel with the replicative helicase during replication. The authors will need to clarify this key point. This may require additional analysis or an additional experiment. Still, I think with revision of this point, the manuscript would be suitable for the EMBO J. Below, I have detailed specific concerns and offer suggestions for addressing them.

Specific Major Concerns:

1. Pg 8: I am not sure I understand what is going on with recruitment.

2. Page 9:

There is strong evidence in this manuscript to support an interaction between the MCM complex and the SHU complex. However, I am less convinced that they are traveling together. One analysis that the authors could perform is to synchronize the cells and to perform a ChIP time course at early and late firing origins sites. Based on the logic used in the manuscript, depletion of SHU from early firing origins should occur faster than from late firing origins, and this should create a characteristic pattern. I think a comparable experiment or re-analysis of the data should be performed to demonstrate this more conclusively.

3. Do Rad51 and Rad52 show similar depletion as replication proceeds?

4. Page 8 and 9: I am not sure I understand the significance of the loss of CSM2 signal at the ARS sequence upon bleomycin, MMS, or HU treatment. What does this mean? I feel like a better explanation of this experiment is needed, and why it was done is needed.

5. Page 11:

How was the human correlation done? This is not explained, and it would help the reader know what is going on.

6. Page 14: Confusing paragraph "While it is tempting to speculate..."

7. Page 17: The Gap filling model is interesting. However, I am not sure there is evidence to support this part of the model. Rad51 could also be recruited to protect regressed forks or other replication challenge structures. I think a more measured interpretation of the model would be better here. We don't really know the specific role of SHU in recruiting Rad51. It would be interesting if it acted as an organizing hub to facilitate all of Rad51's functions at the advancing fork. I guess I think the interpretation should be more measured.

Minor:

Figure 1DE: What are the red lines that look like they are hanging out in space?

Figure 6D- The -MMS control looks a little odd. It might be helpful to repeat this.

Referee #2:

Manuscript:

RAD51 paralogs interact with the replicative helicase to bypass DNA damage

The manuscript by Fagunloye et al. provides interesting and important insight into the link between the Shu complex and replication machinery, which enables error-free bypass of DNA lesions.

Here, authors apply genome-wide ChIP-seq of Shu Complex components and identify that Shu complexes are enriched at the replication origins. Authors propose that Shu complexes are loaded at the replication sites via ORC complexes and travel with the replisome to promote DNA replication stress tolerance. Authors also show that the yeast and human Shu complexes coevolved with ORC and MCM complexes. Using a combination of mutations in Shu Complex components and employing Y2H and Co-IPs, authors establish direct interactions between Shu Complex and the ORC and MCM complexes.

Overall, the study is very interesting, and the experiments are generally well-conducted. The obvious issue with the manuscript is that despite generating several tools and insights from genome-wide data, the authors do not develop the work deeply. There are several open ends that have not been developed, thereby compromising the novelty of the study.

There are several outstanding points that authors could improve and strengthen before publication. Some of the specific points are listed below:

1. In Fig. 1, the authors find that Csm2 is located at the ARS and later conclude that the Shu complex is loaded during the G1 phase by ORCs (Fig 3H) and continues to travel with replisome during the S phase. However, in Fig 1F, they show that the Csm2 protein is largely absent in the G1 phase (20' timepoint). How do ORCs and Csm2 interact in G1 without Csm2 protein in this scenario? To fully understand this result, authors should perform chromatin fractionation from these synchronized cells and properly profile Csm2 and other Shu partners.

The authors state that the graph in Fig 1F is derived from 3 experiments; the author should add error bars in the plot. Despite

authors complaining about the saturated immunoblot signals in their previous study, which led them to conclude that Csm2 is not a cell cycle-regulated protein, authors still present saturated blots in 1G. Authors should properly quantify the levels of Psy2 and Shu1/2 in these blots. Clearly, Psy3 signals increased from 0' to 20' timepoint.

In Fig. 1A and EV1F, authors should present the coordination of ChIP-seq peaks.

2. In Fig 2, authors find very interesting results where they observe DNA damage-induced de-enrichment of Csm2 from ARS sites. This experiment would greatly benefit from controlling pan-chromatin loading of Csm2/Shu components by performing simple chromatin extractions.

Authors argue that Csm2 enrichment could be due to fork movement, but HU, which arrests fork movement, yields the same outcome. In this scenario, could authors deplete replisome progression components or fork remodeling/reversal enzymes and observe what happens to Shu components at ARSs/pan-chromatin upon MMS/HU stress?

3. Referring to Point 1 above, could authors repeat their Co-IPs in Fig 3H from synchronized G1/S/G2 cells?

Again, given the abundance of ORC/MCMs, performing Co-IPs in Fig 3G-I from chromatin inputs would more clearly establish the specificity of these complexes.

Authors have directed their findings on the enrichment of the Shu complex at ARSs and interactions with ORCs to the DNA damage tolerance. However, in light of their new findings (and available mutants), could authors properly analyze replication origin activation, replisome assembly, and cell cycle transition when Shu Complex cannot access ARSs (Csm2-KRRR; Csm2-F46A, Psy3 Y10A)?

Other points:

1. These findings cannot be generalized entirely to canonical and Shu complex RAD51 paralogs; therefore, authors should indicate the Shu complex in the title.

2. Table 1: Plasmids Y2H plasmids nomenclature is expressed with protein and a few with gene. It is good to keep uniformity.

3. The list of antibodies and chemicals appears incomplete.

Referee #3:

In this study, Adeola Fagunloye et al. investigated the interaction they found between the SHU and the ORC and MCM complex. This finding is potentially interesting, but its biological relevance is only partially addressed. The authors were able to identify 5 mutants in the SHU subunit Psy3 that are defective in their interaction with Mcm4 and Orc6. One of them, Psy3-Y10A, shows only a modest sensitivity phenotype when grown on medium containing 0.03% MMS (a very high dose), weaker than the sensitivity observed for a complete PSY3 deletion. It would be prudent to test the phenotype of the other mutants in vivo to confirm this. If this interaction is important for SHU function, and according to the proposed model, i.e. SHU recruitment to the replication machinery is important to avoid DNA lesions through Rad51-dependent template switching, Psy3 loss of interaction mutants should lead to an increase in spontaneous mutagenesis and a lower rate of spontaneous homologous recombination. This could be easily tested.

Other major revisions need to be considered:

The bioinformatics analyses shown in Fig. 1 need to be briefly explained so that the reader can easily understand the data. The same is true for the violin plots in Fig. 3. It is impossible to see any difference in the data presented, except for the different p-values. Please briefly explain the methods and data presentation so that the reader can understand the results.

The Csm2-K189A, R190A, R191A, R192A-6HA mutants used for the ChIP experiments shown in Fig2A should be described. If they have been described previously, please provide the reference and briefly explain how they were designed.

It would be important to know when SHU binds ARS in the cell cycle. A time course version of this experiment starting from a G1 arrested cell population would allow to know if Csm2 is recruited to ARS in G1, S and G2 phases of the cell cycle. In addition, time course experiments should be performed after treatment with MMS and beyond in *apn1Δ apn2Δ* cells. This would help to understand why Csm2 recruitment is decreased after MMS treatment, which contradicts the proposed model where abasic sites should increase SHU recruitment to chromatin. Perhaps MMS treatment leads to cell cycle arrest, which would reduce origin firing and SHU recruitment to ARS? This needs to be tested and properly discussed.

On Fig2D, Csm2-F46A ChIP is lower at ARS than at ChrIV IG, suggesting that Rad55-Rad57 is particularly important for SHU recruitment at ARS, this should be discussed.

It is surprising that Psy3 and Csm2 DNA binding is not required for Mcm/Orc binding (Fig 3J), but for binding to ARS (Fig 2A). It is therefore important to test whether the interaction between Psy3 and Mcm/Orc is required for SHU recruitment to ARS. Same idea with Rad55: interaction with Csm2 is required for binding to ARS (Fig. 2D), but not for SHU-Mcm4 interaction. This should be discussed.

Since only Orc6 seems to show a Y2H interaction with Rad55 in Fig3D, it would be important to confirm that it is not mediated by SHU.

In the Discussion section, it is suggested that SHU travels with the MCM complex. This is an important conclusion that could be tested by additional ChIP-seq experiments. SHU and MCM ChIP performed in synchronized cells released from G1 arrest should show that these two complexes travel together from the ARS during replication. Hydroxyurea could be used to slow down replication and follow the complexes more easily if needed.

If the two complexes do travel together, how do you explain the enrichment of Csm2 at ARS in an exponentially growing population of cells observed by ChIP seq (Fig. 1A)?

ChIP seq seems to be performed with DNA fragments of up to 1 kb, which seems large to provide accurate results. Could you comment on this?

Minor revisions

I could not correct typos due to the lack of page and line numbering.

Figure legends could be simplified, e.g. by explaining the meaning of p-values only once.

Introduction and Discussion: It is misleading to say that SHU has a role in repairing replicative damage, it is involved in DNA damage avoidance.

Rebuttal letter

Fagunloye *et al.* (2024)

We would like to start by sincerely thanking you for taking the time to read and review our manuscript. We deeply appreciate your critical but constructive feedback, which has been instrumental in guiding us through the process of improving our work.

In response to your comments, we have performed several additional experiments, which are now included in both the main and extended figures of the revised manuscript. We have also carefully revised several key sections of the text to address the concerns and suggestions raised in the reviews.

Below, we provide a detailed summary of all the changes made to the figures, including the new data incorporated into the updated version of our manuscript:

Figure number	Former figure number	Breakdown of figure number	Modifications (All figure legends are now simplified)
Figure 1	Unchanged	Figure 1A-E	Updated figure legend for clarity purposes
	Unchanged	Figure 1F-G	Updated the quantification for Csm2 and Psy3 cell cycle analysis
Figure 2	Unchanged	Figure 2A-E	Figure legend simplified
Figure 3	Unchanged	Figure 3A-G, I and J	Figure legend simplified and updated
	Unchanged	Figure 3H (New Data)	co-immunoprecipitation (coIP) experiments of Orc2 with Psy3 and Csm2 in G1 and S/G2 phases
Figure 4	Unchanged	Figure 4A-G	Unchanged
Figure 5	Unchanged	Figure 5C (New data)	Y2H interaction of Rad55 and Orc6 upon PSY3 or CSM2 deletion.
Figure 6	Unchanged	Figure 6D (New data)	MMS sensitivity of psy3 N-terminus mutants in pRS413 vector
	New data	Figure 6E	Canavanine mutational assay of psy3-Y10A
	New data	Figure 6F	Infographic of direct repeat recombination (DRR) assay
	New data	Figure 6G	DRR assay of WT, psy3Δ or psy3-Y10A with and without MMS.
Figure 7	unchanged	Figure 7 (Model)	Figure legend updated for clarity
EV1	unchanged	Figure EV1A-F	Figure legend simplified
EV2	unchanged	Figure EV2A-B	Figure legend simplified
EV3	unchanged	Figure EV3A-E	Figure legend simplified
EV4	unchanged	Figure EV4A	Figure legend simplified
EV5	unchanged	Figure EV5	Figure legend simplified
Appendix S1	New data	Figure S1A	Western blot showing the protein expression of the N-terminus mutant
	New data	Figure S1B	Total recombination rate of WT, psy3Δ or psy3-Y10A with and without MMS.

We believe that the inclusion of these new experiments, along with the critical revisions to the manuscript, has significantly strengthened the overall quality and clarity of the work. We hope that the revised version meets your expectations and we look forward to your further feedback. Once again, we are grateful for your valuable insights and guidance throughout this process.

Response to Reviewer Comments:**Referee #1:****1. Pg 8: I am not sure I understand what is going on with recruitment.**

- We agree with the Reviewer that the text could be clearer, and we have now changed the language from recruitment to enrichment throughout the text.

“Csm2 enrichment to ARS sites depends on its DNA binding activity and interaction with Rad55”- Lines 182-183. (others, Lines 185,195, 201, 202, 207, 211 etc.)

2. Page 9:

There is strong evidence in this manuscript to support an interaction between the MCM complex and the SHU complex. However, I am less convinced that they are traveling together. One analysis that the authors could perform is to synchronize the cells and to perform a ChIP time course at early and late firing origins sites. Based on the logic used in the manuscript, depletion of SHU from early firing origins should occur faster than from late firing origins, and this should create a characteristic pattern. I think a comparable experiment or re-analysis of the data should be performed to demonstrate this more conclusively.

- We fully agree with the referee, and we would love to include these data. However, we encountered multiple problems in synchronizing Csm2-tagged yeast. They arrest with alpha factor nicely, but the release is not in synchrony and varies a lot between replicates. The second strain, which we do in parallel, in which we tagged MCM is proceeding through S phase as expected.
- To overcome these problems, we arrested both cell lines with alpha factor and released the yeast into media containing different HU concentrations. Previously, the Luke lab has used this method to arrest yeast in different phases of the S phase. We successfully, with this method, synchronized both cell lines. The quality of the ChIP and qPCR also looks good, but as we stated in the previous version of our manuscript, DNA damage leads to the uncoupling of Csm2 and MCM, and we do not see an overlap, as expected. We cannot provide good results for the suggested experiments based on these experiments and have therefore softened the language stating that the Shu complex travels with the replicative helicase.

3. Do Rad51 and Rad52 show similar depletion as replication proceeds?

- We agree with the reviewer that while it would indeed be informative to understand if Rad51 and Rad52 exhibit a similar binding pattern with replication progression, this experiment presents significant technical limitations. Notably, both N- and C-terminal tagging of Rad51 has been shown to produce non-functional proteins, complicating any analysis of Rad51's dynamics through ChIP. Additionally, our attempts at tagging Rad52 led to strains with impaired growth, indicating that tagged Rad52 may not behave as it does in untagged conditions, thereby limiting the relevance of such ChIP data. Given

these challenges, we believe that pursuing this experiment would not yield reliable results and is beyond the scope of the current study. Instead, our focus on Csm2 provides a representative and functional insight into the Shu complex's role in DNA damage tolerance, as it interacts with Rad51 and Rad52 in the cellular context.

4. Page 8 and 9: I am not sure I understand the significance of the loss of CSM2 signal at the ARS sequence upon bleomycin, MMS, or HU treatment. What does this mean? I feel like a better explanation of this experiment is needed, and why it was done is needed.

- We agree with the reviewer that a more detailed explanation of the experiment is warranted. We wanted to address whether Csm2 enrichment at ARS sites would be reduced as cells progress through S-phase (which we assessed by arresting cells in S phase with HU) or when DNA damage is induced (using MMS or bleomycin). Indeed, Csm2 enrichment is reduced suggesting that Csm2 is relocalized away from the ARS under these DNA damage conditions and during S phase. We have expanded upon this section in the results.

“These findings suggest a model in which Csm2 moves with the replication fork away from the replication origin during S phase or due to its relocation to alkylation-induced DNA damage at other sites”-Lines 208-210.

“Therefore, these findings suggest that Csm2 loss at ARS sites upon DNA damage exposure is not a general feature of recombination proteins or replication factors”-Lines 218-220.

5. Page 11: How was the human correlation done? This is not explained, and it would help the reader know what is going on.

- We appreciate the Reviewer's feedback that the methodology used for the human evolutionary rate correlation analysis was not adequately explained. We have expanded the discussion of both human and yeast ERC analyses in the Results and provided a detailed explanation of the statistical tests used in the figure legend. In addition, we have provided more in-depth methodological details in the Methods.

“Like yeast, the human Shu complex is also significantly positively correlated with the same complexes (Fig. 3B; CMG helicase, MCM, and ORC complexes). This was determined from a dataset of 63 mammal species. We find that ERC values between human Shu complex proteins also correlated more with members of these 3 complexes than random gene sets (permutation test)”-Lines 267-271.

“Likewise, the mammalian Shu vs complex permutation test was performed on a genome-wide matrix of 19,149 orthologous genes, selecting 1000 random subsets as the null distribution. The code to run ERC can be found at: <https://github.com/nclark-lab/erc>”-Lines 585-588

6. Page 14: Confusing paragraph "While it is tempting to speculate..."

- We agree with the Reviewer that this paragraph is confusing and we apologize for it. We have now separated the results into two paragraphs and re-written this section to be clearer and straight forward.

“Next, we asked if the interaction between the Shu complex and the MCM or ORC complexes may be dependent on one of the Shu complex subunits. We individually knockout out the four Shu complex members in the Y2H strain and found that disruption of PSY3 results in loss of Shu2 and Csm2 interaction with either Mcm4 or Orc6 (Fig. 4E). Consistently, Csm2 enrichment at ARS sites is also reduced in psy3Δ cells (Fig. 4F). Note that loss of PSY3 results in reduced expression of Csm2 and Shu2 (Fig. 4G). Together, these findings indicate that while the Shu complex interacts with ORC and MCM complexes independently of RAD51 and RAD55, the presence of Psy3 is essential for these interactions. Psy3 not only facilitates the physical association of Shu2 and Csm2 with the ORC and MCM complexes but also supports their stability and localization to ARS sites.”-Lines 332-341

7. Page 17: The Gap filling model is interesting. However, I am not sure there is evidence to support this part of the model. Rad51 could also be recruited to protect regressed forks or other replication challenge structures. I think a more measured interpretation of the model would be better here. We don't really know the specific role of SHU in recruiting Rad51. It would be interesting if it acted as an organizing hub to facilitate all of Rad51's functions at the advancing fork. I guess I think the interpretation should be more measured.

- The Reviewer is correct that it is presumed that the function of the Shu complex is to facilitate a template switch that enables the bypass of the lesion through gap filling. While the Shu complex function in promoting Rad51 mediated strand exchange has been explicitly shown (Gaines et al 2015 Nat. Communications, 10:3515. PMID: 26215801), whether or not replication gaps form has not been established. Therefore, we have removed the reference- “gap filling.”

“The Shu complex interacts with Rad52 and Rad51 to enable the bypass of replication damage by template switch (Gaines et al, 2015)”-Lines 345-346.

Minor:

Figure 1DE: What are the red lines that look like they are hanging out in space?

- We apologize for any confusion. The red lines depicted in Figure 1D and 1E are intended to represent the p-value threshold in our ChIP-seq analysis, delineating the overlap of Csm2 at Pol2 sites in Figure 1D and Rad55 at Pol2 sites in Figure 1E. We have now included this explanation within the Figure legend.

“Based on ChIPseq peaks we calculated the overlap ChIPseq binding sites to other genomic features. P-values denote the statistical significance of the enrichment of experimental-determined peaks to random genome sets. The location of the red lines indicates the significance; the further to the right the line is, the lower the calculated p-value is.”-Lines 718-720.

“Overlap of RAD55 peaks with Pol II binding sites with red lines depicting p-value threshold delineating overlap between Rad55 and Pol2 sites”-Lines 726-727

Figure 6D- The -MMS control looks a little odd. It might be helpful to repeat this.

- We thank the reviewer again for their astute suggestion. We have repeated this experiment using the five (5) N-terminal mutants of Psy3, as suggested by Reviewer 3 in Point 1. This result helped to address the concerns raised about the control's appearance as well as providing clarity that the psy3 N-terminal mutants are MMS sensitive.

“By complementing a psy3Δ cell with a plasmid expressing either PSY3-Y10A, -L12A, -F15A, -T17A, or -S18A, we find that all these mutants are unable to fully complement a psy3Δ cell MMS sensitivity in comparison to WT PSY3 (Fig. 6D). We performed western blot to confirm that these N-terminal psy3 mutants are expressed (Fig. EV6A). Note that PSY3-Y10A is in an invariant residue and confers the greatest MMS sensitivity (Fig. 6D). Therefore, we used this mutant as an N-terminal representative for additional studies”-Lines 388-393.

Referee #2:

1. In Fig. 1, the authors find that Csm2 is located at the ARS and later conclude that the Shu complex is loaded during the G1 phase by ORCs (Fig 3H) and continues to travel with replisome during the S phase. However, in Fig 1F, they show that the Csm2 protein is largely absent in the G1 phase (20' timepoint). How do ORCs and Csm2 interact in G1 without Csm2 protein in this scenario? To fully understand this result, authors should perform chromatin fractionation from these synchronized cells and properly profile Csm2 and other Shu partners.

- The reviewer is correct that Csm2 protein levels are reduced, not absent in G1 in comparison to S/G2. In contrast, Psy3 protein levels are constant throughout the cell cycle (Fig. 1F-G). From the IP results presented in **Fig. 3H**, there is less Csm2 relative to Psy3 (in the input and IP lanes), but we are still able to IP Csm2 even in G1 arrested cells. Therefore, it is possible that Orc2 interacts with Csm2 in G1 even though the overall protein levels are reduced. We have now discussed this in the Results section and changed the language to largely reduced in G1.

“We find that Csm2 protein levels are present in G1 although largely reduced during G1 but increase during S/G2 phases, starting at 40 minutes after α-factor release and then reach a plateau (Fig. 1F). In contrast, the G2/M cyclin, Clb2, protein levels peak at 60-80 minutes after α-factor release (Fig. 1F)”-Lines 160-163.

“We next assessed whether Psy3 or Csm2 co-IP endogenous Orc2 in G1 when the ORC complex is loaded onto the replication origins or in S/G2 arrested cells (Fig. 3H). While Psy3 and Csm2 co-IP Orc2 in both G1 and S/G2, Psy3 and Csm2 pull down more Orc2 in the G1 arrested cells.”-Lines 306-309.

- With regards to chromatin fractionation experiments, we have shown that Csm2 and Psy3 DNA binding is largely dispensable for its interaction with members of the MCM or ORC complexes (**Fig. 3I and 3J**). These results are consistent with those from the

Prado laboratory showing that Mcm4 interaction with Rad51 or Rad52 is also DNA independent (Cabello-Lobato *et al* 2021 Cell Reports 36:109440, PMID: 34320356). Furthermore, we have previously shown that the majority of Csm2 is not enriched in the chromatin in the absence of DNA damage even in S phase when looking at bulk DNA (Rosenbaum *et al* 2019 Nature Communications 10:3515; PMID: 31383866 See published Figure below). Csm2 enrichment in chromatin is significantly enriched when abasic sites accumulate upon MMS damage. Therefore, it is unlikely that we will observe an interaction between Csm2 and the MCM or ORC complex in the chromatin fraction because of these two points. It is important to note that the Prado lab found that Mcm4 interacts with Rad51 and Rad52 in a nuclease insoluble fraction (PMID: 34320356).

The authors state that the graph in Fig 1F is derived from 3 experiments; the author should add error bars in the plot. Despite authors complaining about the saturated immunoblot signals in their previous study, which led them to conclude that Csm2 is not a cell cycle-regulated protein, authors still present saturated blots in 1G. Authors should properly quantify the levels of Psy2 and Shu1/2 in these blots. Clearly, Psy3 signals increased from 0' to 20' timepoint.

- We have revised Figures 1F and 1G to have non-saturated blots. We have quantified the protein levels of Psy3, Shu1, and Shu2 and incorporated error bars in the graphs from the three experiments. Regarding the Psy3 signal, the updated quantification shows that Psy3 protein levels remain consistent throughout the time course.

In Fig. 1A and EV1F, authors should present the coordination of ChIP-seq peaks.

- For simplicity, we did not put the coordinates in the figure and have included them in the figure legend. We now include a data availability section that lists all ChIPseq peaks with coordinates defined.

“ChIP-seq data have been uploaded in the National Center for Biotechnology Information (NCBI) Sequencing Read Archive under the reference number PRJNA1112421. All data are available upon request from the corresponding author”-Lines 698-700.

2. In Fig 2, authors find very interesting results where they observe DNA damage-induced de-enrichment of Csm2 from ARS sites. This experiment would greatly benefit from controlling pan-chromatin loading of Csm2/Shu components by performing simple chromatin extractions.

- We thank the referee for this comment. To exclude a general decrease in immunoprecipitation efficiency upon DNA damage (e.g. MMS), we have quantified the overall amount of DNA immunoprecipitated (by ChIP of Csm2) in untreated cells as well as after MMS treatment and find that it is unchanged. We have now included this information.

“We find that Csm2 enrichment at ARS216, ARS306, and ARS305 is reduced in MMS-exposed cells (Fig. 2B). Note, this reduction in binding is not due to overall changes in ChIP efficiency in these strains”-Lines 195-197.

Authors argue that Csm2 enrichment could be due to fork movement, but HU, which arrests fork movement, yields the same outcome. In this scenario, could authors deplete replisome progression components or fork remodeling/reversal enzymes and observe

what happens to Shu components at ARSs/pan-chromatin upon MMS/HU stress?

- We agree with the Reviewer that this is a fantastic idea, but it is beyond the scope of this manuscript.

3. Referring to Point 1 above, could authors repeat their Co-IPs in Fig 3H from synchronized G1/S/G2 cells?

- We appreciate this suggestion. We have performed co-immunoprecipitation (coIP) experiments from cells synchronized in both the G1 (alpha factor arrested) and S/G2 (released from alpha factor for 50 min) phases. We find that Psy3 and Csm2 coIP Orc2 in both G1 and S/G2. However, we find that Psy3 and Csm2 pull down more Orc2 in the G1 arrested cells.

New Figure 3H:
Figure 3G Legend: Psy3 and Csm2 co-IP ORC2 during G1 and S/G2 phase cells. Untagged wild-type, Psy3-6HA, or Csm2-6HA expressing cells were arrested in G1 with α -factor or released from α -factor into S/G2. Psy3 and Csm2 was IP using HA antibodies and then run on an SDS-PAGE gel. Co-IP with ORC2 was accessed using anti-ORC2 antibodies and immunoprecipitation was accessed using anti-HA antibodies. The input represents 5% of the total.

“We next assessed whether Psy3 or Csm2 co-IP endogenous Orc2 in G1 when the ORC complex is loaded onto the replication origins or in S/G2 arrested cells (Fig. 3H). While Psy3 and Csm2 co-IP Orc2 in both G1 and S/G2, Psy3 and Csm2 pull down more Orc2 in the G1 arrested cells.”-Lines 306-309

Again, given the abundance of ORC/MCMs, performing Co-IPs in Fig 3G-I from chromatin inputs would more clearly establish the specificity of these complexes.

- As mentioned above and with regards to chromatin fractionation experiments, we have shown that Csm2 and Psy3 DNA binding is largely dispensable for its interaction with members of the MCM or ORC complexes (**Fig. 3I and 3J**). These results are consistent with those from the Prado laboratory showing that Mcm4 interaction with Rad51 or Rad52 is also DNA independent (Cabello-Lobato *et al* 2021 Cell Reports 36:109440, PMID: 34320356). Furthermore, we have previously shown that the majority of Csm2 is not enriched in the chromatin in the absence of DNA damage even in S phase when looking at bulk DNA (Rosenbaum *et al* 2019 Nature Communications 10:3515; PMID: 31383866). Therefore, it is unlikely that we will observe an interaction between Csm2 and the MCM or ORC complex in the chromatin fraction.

Authors have directed their findings on the enrichment of the Shu complex at ARSs and interactions with ORCs to the DNA damage tolerance. However, in light of their new findings (and available mutants), could authors properly analyze replication origin

activation, replisome assembly, and cell cycle transition when Shu Complex cannot access ARSs (Csm2-KRRR; Csm2-F46A, Psy3 Y10A)?

- We thank the reviewer for these suggestions and the analysis of replication origin activation, replisome assembly and cell cycle transition are all important aspects to consider. However, these studies are beyond the scope of the current manuscript. In the absence of the Shu complex, it has already been shown that there is a mild defect in S phase progression in the presence of MMS (Mankouri *et al* 2007 *Molecular Biology of the Cell*, 18; 4062, PMID: 17671161). This result suggests that it is highly unlikely that the Shu complex will alter replication origin activation or replisome assembly, but the Shu complex does have a profound impact on error-free repair. While it is likely that the *psy3-Y10A* mutant may also lead to a similar S phase defect, the results are difficult to discern in the bulk assays. We have now discussed these points as a future direction.

“In the future, it will be important to determine if the Shu complex influences replication origin activation or replisome assembly and to delineate how the Shu complex moves or travels with the replication machinery”-Lines 481-483

Other points:**1. These findings cannot be generalized entirely to canonical and Shu complex RAD51 paralogs; therefore, authors should indicate the Shu complex in the title.**

- We appreciate and agree with this suggestion. To comply with the reviewer’s request, we have now changed the title to: “The Shu complex interacts with the replicative helicase to bypass DNA damage”.

2. Table 1: Plasmids Y2H plasmids nomenclature is expressed with protein and a few with gene. It is good to keep uniformity.

- We have now revised Table 1 to ensure that all entries uniformly use either protein names or gene names, as appropriate, to maintain consistency.

3. The list of antibodies and chemicals appears incomplete.

- We have updated the list of antibodies and chemicals used throughout our experiments.

Referee #3:

In this study, Adeola Fagunloye et al. investigated the interaction they found between the SHU and the ORC and MCM complex. This finding is potentially interesting, but its biological relevance is only partially addressed. The authors were able to identify 5 mutants in the SHU subunit Psy3 that are defective in their interaction with Mcm4 and Orc6. One of them, Psy3-Y10A, shows only a modest sensitivity phenotype when grown on medium containing 0.03% MMS (a very high dose), weaker than the sensitivity observed for a complete PSY3 deletion. It would be prudent to test the phenotype of the other mutants in vivo to confirm this. If this interaction is important for SHU function, and according to the proposed model, i.e. SHU recruitment to the replication machinery is important to avoid DNA lesions through Rad51-dependent template switching, Psy3 loss of interaction

mutants should lead to an increase in spontaneous mutagenesis and a lower rate of spontaneous homologous recombination. This could be easily tested.

- We appreciate the reviewer's positive remarks, we have extended our studies to include an MMS sensitivity assay for all five N-terminal mutants of *psy3*, as well as performed mutagenesis and direct repeat recombination assays for the invariant *Psy3-Y10A* mutant. These experiments clearly demonstrate that the interaction between *Psy3* and the replication machinery is critical for suppressing mutations and error-prone repair.

NEW ADDITION- The conserved N-terminus of *Psy3* mediates Shu complex interaction with *Mcm4* and *Orc6*. (new figure 6D-G below, and additional data in EV6)

The bioinformatics analyses shown in Fig. 1 need to be briefly explained so that the reader can easily understand the data. The same is true for the violin plots in Fig. 3. It is impossible to see any difference in the data presented, except for the different p-values.

Please briefly explain the methods and data presentation so that the reader can understand the results.

- We agree with the reviewer that clearer explanations of the bioinformatics analyses in Figure 1 and the violin plots in Figure 3 are warranted. We have updated the manuscript to include a more detailed explanation of these methods in the Results.

The Csm2-K189A, R190A, R191A, R192A-6HA mutants used for the ChIP experiments shown in Fig 2A should be described. If they have been described previously, please provide the reference and briefly explain how they were designed.

- The reviewer is correct that the *Csm2-K189A*, *R190A*, *R191A*, and *R192A-6HA* mutants were previously described in Rosenbaum et al 2019 Nature Communications 10:3515, PMID: PMC6683157. We have now included this reference to depict how they were integrated which was at the endogenous locus and promoter with expression verified by western blot.

It would be important to know when SHU binds ARS in the cell cycle. A time course version of this experiment starting from a G1 arrested cell population would allow to know if Csm2 is recruited to ARS in G1, S and G2 phases of the cell cycle.

- As mentioned in our response to Referee 1, we would love to include these data. However, we encountered multiple problems in synchronizing Csm2-tagged yeast. They arrest nicely with alpha factor, but the release is not in sync and varies a lot between replicates. The second strain, which we do in parallel and in which we tagged MCM, is proceeding through the S phase as expected.
- To overcome these problems, we arrested both cell lines with alpha factor and released the yeast into media containing different HU concentrations. Previously, the Luke lab has used this method to arrest yeast in different phases of the S phase. We successfully, with this method, synchronized both cell lines. The quality of the ChIP and qPCR also looks good, but as we stated in the previous version of our manuscript, DNA damage leads to the uncoupling of Csm2 and MCM, and we do not see an overlap, as expected. We cannot provide good results for the suggested experiments based on these experiments. However, in the current manuscript, we have now included strong findings providing further evidence that SHU is traveling together with MCM. However, as the direct proof is missing, we discuss this aspect in the discussion.

In addition, time course experiments should be performed after treatment with MMS and beyond in *apn1Δ apn2Δ* cells. This would help to understand why Csm2 recruitment is decreased after MMS treatment, which contradicts the proposed model where abasic sites should increase SHU recruitment to chromatin. Perhaps MMS treatment leads to cell cycle arrest, which would reduce origin firing and SHU recruitment to ARS? This needs to be tested and properly discussed.

- We previously showed that the majority of Csm2 is not enriched in the chromatin in the absence of DNA damage even in S phase when looking at bulk DNA (See published Figure below; Rosenbaum et al 2019 Nature Communications 10:3515, PMID: PMC6683157). Csm2 enrichment in chromatin is significantly enriched when abasic

sites accumulate upon MMS damage (See published Figure below; Rosenbaum et al 2019 Nature Communications 10:3515, PMID: PMC6683157). We hypothesize that when abasic sites accumulate then the Shu complex binds to the DNA with greater affinity. We have previously shown that the DNA binding subunits of the Shu complex, Csm2-Psy3, have greater affinity for a double flap substrate containing an abasic site mimetic (Rosenbaum et al 2019 Nature Communications 10:3515, PMID: PMC6683157).

We agree with the reviewer that MMS's treatment leads to cell cycle arrest (see Figure below), which could reduce origin firing. However, we observe equal cell cycle arrest for all the strains tested and therefore origin firing is unlikely to be different. Note that the Prado lab also performed an MMS time course examining cell cycle progression by FACS (González-Prieto *et al* 2013 EMBO J., 32:1307, PMID: PMC3642682).

W303 strain untreated.

W303 strain after 2 hours 0,02% MMS's treatment

In Fig 2D, Csm2-F46A ChIP is lower at ARS than at ChrIV IG, suggesting that Rad55-Rad57 is particularly important for SHU recruitment at ARS, this should be discussed.

- We agree with the Reviewer and have now discussed how Rad55-Rad57 plays an important role for Shu complex recruitment to ARS sites.

“Intriguingly, Csm2 enrichment at ARS sites is largely dependent on its interaction with Rad55. Interestingly, Rad55 is needed for Csm2 enrichment at ARS sites while being dispensable for Shu complex interaction with Mcm4. These results are consistent with those from the Prado laboratory showing that Mcm4 interaction with Rad51 or Rad52 is also DNA-independent (Cabello-Lobato et al, 2021). Furthermore, we show that Csm2 and Psy3 DNA binding is largely dispensable for its interaction with members of the MCM or ORC complexes (Fig. 3I and 3J). Therefore, it is possible that Rad55 helps to stabilize or enrich the Shu complex to ARS sites but that the Shu complex alone is needed to interact with the replisome”-Lines 472-479.

It is surprising that Psy3 and Csm2 DNA binding is not required for Mcm/Orc binding (Fig 3J), but for binding to ARS (Fig 2A). It is therefore important to test whether the interaction between Psy3 and Mcm/Orc is required for SHU recruitment to ARS.

- We agree with the Reviewer that it is important to test if the interaction between Psy3 and Mcm/Orc is important for its recruitment to ARS sites. However, we encountered significant technical challenges in integrating this mutant which is needed to study this interaction under its endogenous promoter and expression. Unfortunately, attempts to create this mutant were unsuccessful in both the Bernstein and Paeschke labs.

Same idea with Rad55: interaction with Csm2 is required for binding to ARS (Fig. 2D), but not for SHU-Mcm4 interaction. This should be discussed.

- We agree with the Reviewer that it is interesting that Rad55 is needed for Csm2 enrichment at ARS sites while being dispensable for Shu complex interaction with Mcm4. These results are consistent with those from the Prado laboratory showing that Mcm4 interaction with Rad51 or Rad52 is also DNA independent (Cabello-Lobato et al 2021 Cell Reports 36:109440). Furthermore, we showed that Csm2 and Psy3 DNA binding is largely dispensable for its interaction with members of the MCM or ORC complexes (**Fig. 3I and 3J**). Therefore, it is possible that Rad55 helps to stabilize or enrich the Shu complex to ARS sites but that the Shu complex alone is needed to interact with the replisome. We have added this explanation to the Discussion.

“Intriguingly, Csm2 enrichment at ARS sites is largely dependent on its interaction with Rad55. Interestingly, Rad55 is needed for Csm2 enrichment at ARS sites while being dispensable for Shu complex interaction with Mcm4. These results are consistent with those from the Prado laboratory showing that Mcm4 interaction with Rad51 or Rad52 is also DNA-independent (Cabello-Lobato et al, 2021). Furthermore, we show that Csm2 and Psy3 DNA binding is largely dispensable for its interaction with members of the MCM or ORC complexes (Fig. 3I and 3J). Therefore, it is possible that Rad55 helps to stabilize or enrich the Shu complex to ARS sites but that the Shu complex alone is needed to interact with the replisome”-Lines 472-479.

Since only Orc6 seems to show a Y2H interaction with Rad55 in Fig3D, it would be important to confirm that it is not mediated by SHU.

- We have now performed the Y2H experiment to examine whether Rad55 and Orc6 interaction depends on the Shu complex members, Csm2 or Psy3. We find that loss of *PSY3*, but not *CSM2*, results in a reduced interaction between Rad55 and Orc6.

New Figure 5C:

Figure 5C Legend: Rad55 Y2H interaction with Orc6 is reduced upon *PSY3* deletion. Y2H experiment examining Rad55 interaction with Orc6 transformed in Y2H strain with either *psy3Δ* or *csm2Δ* knocked out. Yeast with the indicated plasmids were grown in SC-L-W, plated on SC-L-W-H medium, and incubated for 2-3 days at 30°C. Empty vectors are negative controls. Growth is indicative of a Y2H interaction and SC-L-W is used as a loading control. All experiments were done in triplicate.

“Since we find in Fig. 3D that Orc6 exhibits a Y2H interaction with Rad55, we examined whether Rad55 and Orc6 interaction depends on the Shu complex members, Csm2 or Psy3. Our result suggests that the loss of PSY3, but not CSM2, may modestly reduce the interaction between Rad55 and Orc6 (Fig. 5C), suggesting that Psy3 plays a subtle but important role in facilitating the Rad55-Orc6 interaction”-Lines 358-362.

In the Discussion section, it is suggested that SHU travels with the MCM complex. This is an important conclusion that could be tested by additional ChIP-seq experiments. SHU and MCM ChIP performed in synchronized cells released from G1 arrest should show that these two complexes travel together from the ARS during replication. Hydroxyurea could be used to slow down replication and follow the complexes more easily if needed.

- As part of our revised experimental approach, we initially planned to perform the suggested experiment with hydroxyurea (HU) treatment to investigate the interactions between MCM and Shu complex members. However, as demonstrated in this manuscript, HU treatment results in the uncoupling of MCM and Shu complex binding, preventing co-localization under these conditions. Consistent with this observation, our ChIP-seq analysis of asynchronous cells, prior to HU treatment, shows a significant overlap between MCM and Shu binding sites, supporting the interaction in unstressed conditions. We have addressed this discrepancy and its implications for our findings in the Discussion section.

If the two complexes do travel together, how do you explain the enrichment of Csm2 at ARS in an exponentially growing population of cells observed by ChIP seq (Fig.

1A)?

- We thank the reviewer regarding the observed enrichment of Csm2 at ARS during the exponential growth of cells as shown in ChIP-seq data from Figure 1A. In an exponentially growing population, cells are distributed across various stages of the cell cycle, with a significant proportion likely being in the S-phase where replication origins (ARS) are most active. This mixed population could result in varied binding interactions visible in our ChIP-seq analysis.

The ChIP-seq technique, while extremely useful for detecting protein-DNA interactions, is not inherently quantitative across the entire genome or all potential binding sites (ARSs). It is sensitive to strong interactions where the protein-DNA binding is more stable and thus more likely to be captured during the immunoprecipitation process. The PCR amplification step in ChIP-seq can disproportionately amplify these interactions, making it challenging to quantify how many cells in the population exhibit the interaction at any given time.

ChIP seq seems to be performed with DNA fragments of up to 1 kb, which seems large to provide accurate results. Could you comment on this?

- For each ChIP-seq sample, we ensured that the DNA fragments were around 200 bp. This was verified both before and after library preparation by gel electrophoresis and Tape Station analysis. This smaller fragment size is ideal for accurately identifying the locations of protein-DNA interactions. Additionally, it is important to note that the peaks identified in ChIP-seq data often represent a cluster of nearby binding sites rather than a single point of interaction. This clustering can give the impression of larger fragments but is a collection of multiple close interaction sites that are captured together due to the nature of the DNA-protein binding and the resolution of the sequencing technology (See Figure).

Figure legends could be simplified, e.g. by explaining the meaning of p-values only once.

- We appreciate the Reviewer's astute observation. We have now simplified the Figure Legends.

Introduction and Discussion: It is misleading to say that SHU has a role in repairing replicative damage, it is involved in DNA damage avoidance.

- We have changed the wording from "repairing replicative damage" to "bypass of replicative damage" due to its function during replication.

"Consistent with a role in the bypass of replicative damage, we also find that Csm2 (Fig. 1D), and Rad55 (Fig. 1E) are significantly enriched at DNA regions where replication stalls/slow as indicated by high levels of DNA Polymerase 2 (DNA Pol2) occupancy ($p < 0.0001$)-Lines 145-148.

Prof. Kara A Bernstein
University of Pennsylvania School of Medicine
Biochemistry and Biophysics
421 Curie Boulevard
BRB II/III Room 411
Philadelphia, PA 19104-6160

3rd Dec 2024

Re: EMBOJ-2024-117270R
The Shu complex interacts with the replicative helicase to bypass DNA damage

Dear Kara and Katrin,

Thank you again for submitting your revised manuscript to The EMBO Journal. All three original referees have now re-reviewed it, and provided the comments copied below. While referees 1 & 2 have only minor points left for modification, referee 3 is still not fully convinced that your results have decisively demonstrated that SHU complex members may travel along with the replicative helicase. I further discussed these concerns with the other two reviewers, who did not dispute referee 3's assessment, but also felt that additional experiments such as those requested may not necessarily resolve these issues in a straightforward manner, and may well require in-depth follow-up in a future study. Nevertheless, both referees 1 and 2 agreed that more balanced interpretation/discussion would be needed, acknowledging that certain aspects remain open and subject to future work (please see excerpts from their cross-comments appended after the reports below).

I am therefore inviting you to revise the writing/presentation of the manuscript to incorporate these assessments, as well as the following editorial issues:

- Please adjust the order of the manuscript sections: Title page with complete author information, Abstract, Keywords, Introduction, Results, Discussion, Methods, Data Availability, Acknowledgements, Disclosure and Competing Interests Statement, References, Main Figure Legends, Tables, Expanded Figure Legends.
- Please shorten the abstract in the text document in accordance with our author guidelines. Also, please consider if the title could be made slightly more explicit to attract a wide readership.
- On the abstract page of the manuscript, please include 4-5 general keyword terms to enhance searchability.
- Please include a "Disclosure and competing interests statement" (next to the Acknowledgment section) - for details, see <https://www.embopress.org/competing-interests>
- Please include the EV Figure legends in the main text, following after the main figure legends.
- Our pre-acceptance text similarity checks found several instances of Material & Methods section passages (e.g., ChIP, ChIP-seq, statistical analysis, flow cytometry) being near-verbatim copies from previous papers by yourself or others. While this is acceptable in principle in the interest of reproducibility, please however make sure to cite the respective earlier works on those occasions ("...was done essentially as in xxx; in particular, ..."). Moreover, parts of the abstract also appear to have been published like this before, and therefore need to be re-written.
- Please rename the bibliography into References, and for the preprint citations (e.g. Little et al), please adjust the citation format as specified in our author guidelines for preprints:
The citation in the text should be: "(preprint: NAME1 et al, YEAR)"
The citation in the reference list: "Author NAME1, Author NAME2, ... (YEAR) article title. bioRxiv/ResearchSquare doi: XXX"
- In the Data Availability section, please include a URL to the database in which the deposited datasets can be accessed.
- Please check for congruency of the funding information listed in the manuscript and entered in our submission system; missing in the system: "Research in the Paeschke laboratory is funded by...CANTAR. The project "CANTAR" is receiving funding from the programme "Netzwerke 2021", an initiative of the Ministry of Culture and Science of the State of Northrhine Westphalia. Further, this project is supported by a grant from the Fritz Thyssen Foundation to KP" => none of these funders are included in eJP, not sure which should be listed; grant number EXC2151 - 390873048 in the submission system is missing the funder name
- Please pre-face the appendix PDF with the line "Appendix for", and make sure to provide each Appendix Figure legend right below the respective Appendix figure.

- Source data files need to be saved according to a scheme: one figure/one folder and then uploaded as .zip files. E.g. all the Source data files for figure 1 need to be saved in a single folder and this needs to be zipped and then uploaded as "SD figure 1.zip" file. For EV and/or appendix figures, ZIP together all source data.

- Finally, during routine pre-acceptance checks, our data editors have raised the following queries regarding figures, data, and legends; I would appreciate if you briefly answered to them in the cover letter of your final submission, and made the requested text modifications with changes/additions highlighted via the "Track changes" option, to facilitate our final checking.

1. Please define the annotated p values *** as well as provide the exact p-values for the same in the legend of figure 6G as appropriate.
2. Please note that the exact p values are not provided in the legends of figures 1B-E, F; 2A-E; 3A, 4F, 6E, G; EV1 B, D, E; EV3 A, B; supplementary figure 1B
3. Please indicate the statistical test used for data analysis in the legends of figures 1B-E, F; 3A, B; 6E, G; EV1 B-E; supplementary figure 1B
4. Please note that information related to n is missing in the legends of figures 3A, B; 6E.
5. Please note that the error bars are not defined in the legend of figure 1G.
6. Please note that the measure of center for the error bars needs to be defined in the legend of figure 6E.

Please do not hesitate to contact me should you have any questions regarding these remaining presentational revisions. I look forward to receiving your final version!

With kind regards,

Hartmut

7) All authors listed as (co-)corresponding need to deposit, in their respective author profiles in our submission system, a unique

ORCID identifier linked to their name. Please see our Guide to Authors for detailed instructions.

9) To facilitate reproducibility and cross-laboratory adoption of methodologies, please structure the Materials & Methods section as outlined in our guide to authors, including a completed Reagents and Tools Table that can be downloaded from our author guidelines as well (<https://www.embopress.org/page/journal/14602075/authorguide#structuredmethods>).

10) Digital image enhancement is acceptable practice, as long as it accurately represents the original data and conforms to community standards. If a figure has been subjected to significant electronic manipulation, this must be clearly noted in the figure legend and/or the 'Materials and Methods' section. The editors reserve the right to request original versions of figures and the original images that were used to assemble the figure. Finally, we generally encourage uploading of numerical as well as gel/blot image source data; for details see: embopress.org/page/journal/14602075/authorguide#sourcedata

At EMBO Press, we ask authors to provide source data for the main manuscript figures. Our source data coordinator will contact you to discuss which figure panels we would need source data for and will also provide you with helpful tips on how to upload and organize the files.

In the interest of ensuring the conceptual advance provided by the work, we recommend submitting a revision within 3 months (3rd Mar 2025). Please discuss the revision progress ahead of this time with the editor if you require more time to complete the revisions. Use the link below to submit your revision:

Link Not Available

Referee #1:

The authors have addressed all of my previous concerns and have added additional data that significantly strengthens the impact and quality of the manuscript. I have one minor comment below. However, I think it is an exciting paper and is suitable for publication in EMBOJ.

Major concerns: None

Minor concerns: The first paragraph in the results section. I think it should be sites and not sides.

Referee #2:

The Shu complex interacts with the replicative helicase to bypass DNA damage.

The authors have done great work addressing most of the critical concerns while clearly and thoughtfully justifying why some of the suggestions extend beyond the scope of the current manuscript.

The inclusion of additional control experiments has indeed significantly strengthened their conclusions and increased the overall robustness of the findings.

That said, however, the significance of CSM2 loss from ARS in response to various DNA-damaging agents remains unclear. This is particularly notable in the context of DNA lesion bypass and the observed increase in CSM2 loading on chromatin following MMS treatment. A more thorough interpretation and expanded discussion on this point would be valuable.

Overall, the findings are interesting and important, and the experiments are well-executed and have strong statistical support. In my opinion, this manuscript merits publication in the EMBO Journal.

Referee #3:

The authors have made efforts to improve their manuscript. In particular, I really appreciate the addition of mutagenesis and direct repeat recombination assays used to study the effect of the psy3 interaction mutant with Mcm and Orc. However, I still think there is not enough data to support the hypothesis that SHU travels with Mcm and Orc from the onset of DNA replication, which weakens the quality of the paper. For the sake of clarity, I decided to write my new review based on the authors' responses to my initial questions.

Referee #3:

In this study, Adeola Fagunloye et al. investigated the interaction they found between the SHU and the ORC and MCM complex. This finding is potentially interesting, but its biological relevance is only partially addressed. The authors were able to identify 5 mutants in the SHU subunit Psy3 that are defective in their interaction with Mcm4 and Orc6. One of them, Psy3-Y10A, shows only a modest sensitivity phenotype when grown on medium containing 0.03% MMS (a very high dose), weaker than the sensitivity observed for a complete PSY3 deletion. It would be prudent to test the phenotype of the other mutants in vivo to confirm this. If this interaction is important for SHU function, and according to the proposed model, i.e. SHU recruitment to the replication machinery is important to avoid DNA lesions through Rad51-dependent template switching, Psy3 loss of interaction mutants should lead to an increase in spontaneous mutagenesis and a lower rate of spontaneous homologous recombination. This could be easily tested

We appreciate the reviewer's positive remarks, we have extended our studies to include an MMS sensitivity assay for all five N-terminal mutants of psy3, as well as performed mutagenesis and direct repeat recombination assays for the invariant Psy3-Y10A mutant. These experiments clearly demonstrate that the interaction between Psy3 and the replication machinery is critical for suppressing mutations and error-prone repair.

This is indeed a very nice addition.

The bioinformatics analyses shown in Fig. 1 need to be briefly explained so that the reader can easily understand the data. The same is true for the violin plots in Fig. 3. It is impossible to see any difference in the data presented, except for the different p-values. Please briefly explain the methods and data presentation so that the reader can understand the results.

We agree with the reviewer that clearer explanations of the bioinformatics analyses in Figure 1 and the violin plots in Figure 3 are warranted. We have updated the manuscript to include a more detailed explanation of these methods in the Results.

It is better now, thanks, but looking at the violin plot in Fig.3, I still have the impression that CMG and GINS do not look that different, while the p-values do. This is not well explained. Maybe you should add numbers to the text for comparison?

The Csm2-K189A, R190A, R191A, R192A-6HA mutants used for the ChIP experiments shown in Fig 2A should be described. If they have been described previously, please provide the reference and briefly explain how they were designed.

The reviewer is correct that the Csm2-K189A, R190A, R191A, and R192A-6HA mutants were previously described in Rosenbaum et al 2019 Nature Communications 10:3515, PMID: PMC6683157. We have now included this reference to depict how they were integrated which was at the endogenous locus and promoter with expression verified by western blot.

ok

It would be important to know when SHU binds ARS in the cell cycle. A time course version of this experiment starting from a G1 arrested cell population would allow to know if Csm2 is recruited to ARS in G1, S and G2 phases of the cell cycle.

As mentioned in our response to Referee 1, we would love to include these data. However, we encountered multiple problems in synchronizing Csm2-tagged yeast. They arrest nicely with alpha factor, but the release is not in sync and varies a lot between replicates. The second strain, which we do in parallel and in which we tagged MCM, is

proceeding through the S phase as expected.

I don't understand this because an alpha factor arrest of Csm2-6HA cells released in G1 and followed in G2 is shown in Fig.1F to study the variation of the amount of Csm2-6HA. You can also use Psy3-6HA or Shu2-6HA. If there is a problem with synchrony, why don't you just do the experiment in alpha-factor arrested cells and nocodazole arrested cells (or with a mutation like *cdc28-as1*) as you did for your co-IP experiments (Fig.3G H I)? I think this experiment is very important to defend your traveling model. What if it turns out that SHU is enriched at ARS in G2?

To overcome these problems, we arrested both cell lines with alpha factor and released the yeast into media containing different HU concentrations. Previously, the Luke lab has used this method to arrest yeast in different phases of the S phase. We successfully, with this method, synchronized both cell lines. The quality of the ChIP and qPCR also looks good, but as we stated in the previous version of our manuscript, DNA damage leads to the uncoupling of Csm2 and MCM, and we do not see an overlap, as expected. We cannot provide good results for the suggested experiments based on these experiments. However, in the current manuscript, we have now included strong findings providing further evidence that SHU is traveling together with MCM. However, as the direct proof is missing, we discuss this aspect in the discussion.

Unfortunately, I don't think there is much evidence that SHU travels with the replication forks. In fact, I think that the fact that Csm2 is more highly expressed in S/G2 suggests that there might be "new" binding of free SHU to Mcm or Orc that is not associated with Mcm and Orc from the start of replication. In addition, the fact that the Csm2 Ip signal decreases at ARS after HU treatment, while forks are probably very slow (actually I have not found the concentration used in this experiment), does not fit with a permanent association of SHU with Mcm. It is also strange in the hypothesis that SHU and Mcm travel together that Mcm ChIP at ARS does not decrease after HU treatment (Fig. EV3D). ChIP or ChIP-Seq experiments in the different phase of the cell cycle and co-localization with Mcm/Orc is required.

In addition, time course experiments should be performed after treatment with MMS and beyond in *apn1Δ apn2Δ* cells. This would help to understand why Csm2 recruitment is decreased after MMS treatment, which contradicts the proposed model where abasic sites should increase SHU recruitment to chromatin. Perhaps MMS treatment leads to cell cycle arrest, which would reduce origin firing and SHU recruitment to ARS? This needs to be tested and properly discussed.

We previously showed that the majority of Csm2 is not enriched in the chromatin in the absence of DNA damage even in S phase when looking at bulk DNA (See published Figure below; Rosenbaum et al 2019 Nature Communications 10:3515, PMID: PMC6683157). Csm2 enrichment in chromatin is significantly enriched when abasic sites accumulate upon MMS damage (See published Figure below; Rosenbaum et al 2019 Nature Communications 10:3515, PMID: PMC6683157). We hypothesize that when abasic sites accumulate then the Shu complex binds to the DNA with greater affinity. We have previously shown that the DNA binding subunits of the Shu complex, Csm2-Psy3, have greater affinity for a double flap substrate containing an abasic site mimetic (Rosenbaum et al 2019 Nature Communications 10:3515, PMID: PMC6683157).

Again, a direct link between Csm2 traveling with Mcm after MMS treatment is lacking. It is still not shown why Csm2 is less recruited to ARS after HU and MMS treatment. In fact, Mcm is not less recruited after HU treatment (Fig. EV3D).

We agree with the reviewer that MMS's treatment leads to cell cycle arrest (see Figure below), which could reduce origin firing. However, we observe equal cell cycle arrest for all the strains tested and therefore origin firing is unlikely to be different. Note that the Prado lab also performed an MMS time course examining cell cycle progression by FACS (González-Prieto et al 2013 EMBO J., 32:1307, PMID: PMC3642682).

ok

In Fig 2D, Csm2-F46A ChIP is lower at ARS than at ChrIV IG, suggesting that Rad55-Rad57 is particularly important for SHU recruitment at ARS, this should be discussed.

We agree with the Reviewer and have now discussed how Rad55-Rad57 plays an important role for Shu complex recruitment to ARS sites.

"Intriguingly, Csm2 enrichment at ARS sites is largely dependent on its interaction with Rad55. Interestingly, Rad55 is needed for Csm2 enrichment at ARS sites while being dispensable for Shu complex interaction with Mcm4. These results are consistent with

those from the Prado laboratory showing that Mcm4 interaction with Rad51 or Rad52 is also DNA-independent (Cabello- Lobato et al, 2021). Furthermore, we show that Csm2 and Psy3 DNA binding is largely dispensable for its interaction with members of the MCM or ORC complexes (Fig. 3I and 3J). Therefore, it is possible that Rad55 helps to stabilize or enrich the Shu complex to ARS sites but that the Shu complex alone is needed to interact with the replisome"-Lines 472-479.

ok

It is surprising that Psy3 and Csm2 DNA binding is not required for Mcm/Orc binding (Fig 3J), but for binding to ARS (Fig 2A). It is therefore important to test whether the interaction between Psy3 and Mcm/Orc is required for SHU recruitment to ARS.

We agree with the Reviewer that it is important to test if the interaction between Psy3 and Mcm/Orc is important for its recruitment to ARS sites. However, we encountered significant technical challenges in integrating this mutant which is needed to study this interaction under its endogenous promoter and expression. Unfortunately, attempts to create this mutant were unsuccessful in both the Bernstein and Paeschke labs.

I don't understand this. It was possible to make the csm2-KRRRR strain and do ChIP experiments. There are 5 Psy3 mutations that could be used to do this experiment in a Psy3, shu2 or Csm2 tagged strain. The Psy3 mutations don't seem to have a negative effect on growth in the Y2H system or in the CAN1 and direct repeat strains. This is an important experiment and should be pursued.

Same idea with Rad55: interaction with Csm2 is required for binding to ARS (Fig. 2D), but not for SHU-Mcm4 interaction. This should be discussed.

We agree with the Reviewer that it is interesting that Rad55 is needed for Csm2 enrichment at ARS sites while being dispensable for Shu complex interaction with Mcm4. These results are consistent with those from the Prado laboratory showing that Mcm4 interaction with Rad51 or Rad52 is also DNA independent (Cabello-Lobato et al 2021 Cell Reports 36:109440). Furthermore, we showed that Csm2 and Psy3 DNA binding is largely dispensable for its interaction with members of the MCM or ORC complexes (Fig. 3I and 3J). Therefore, it is possible that Rad55 helps to stabilize or enrich the Shu complex to ARS sites but that the Shu complex alone is needed to interact with the replisome. We have added this explanation to the Discussion.

"Intriguingly, Csm2 enrichment at ARS sites is largely dependent on its interaction with Rad55. Interestingly, Rad55 is needed for Csm2 enrichment at ARS sites while being dispensable for Shu complex interaction with Mcm4. These results are consistent with those from the Prado laboratory showing that Mcm4 interaction with Rad51 or Rad52 is also DNA-independent (Cabello- Lobato et al, 2021). Furthermore, we show that Csm2 and Psy3 DNA binding is largely dispensable for its interaction with members of the MCM or ORC complexes (Fig. 3I and 3J). Therefore, it is possible that Rad55 helps to stabilize or enrich the Shu complex to ARS sites but that the Shu complex alone is needed to interact with the replisome"-Lines 472-479.

Ok

Since only Orc6 seems to show a Y2H interaction with Rad55 in Fig3D, it would be important to confirm that it is not mediated by SHU.

We have now performed the Y2H experiment to examine whether Rad55 and Orc6 interaction depends on the Shu complex members, Csm2 or Psy3. We find that loss of PSY3, but not CSM2, results in a reduced interaction between Rad55 and Orc6.

Ok, but I wonder if this new paragraph would be better placed in the first description of Fig. 3D.

In the Discussion section, it is suggested that SHU travels with the MCM complex. This is an important conclusion that could be tested by additional ChIP-seq experiments. SHU and MCM ChIP performed in synchronized cells released from G1 arrest should show that these two complexes travel together from the ARS during replication. Hydroxyurea could be used to slow down replication and follow the complexes more easily if needed.

As part of our revised experimental approach, we initially planned to perform the

suggested experiment with hydroxyurea (HU) treatment to investigate the interactions between MCM and Shu complex members. However, as demonstrated in this manuscript, HU treatment results in the uncoupling of MCM and Shu complex binding, preventing co-localization under these conditions. Consistent with this observation, our ChIP-seq analysis of asynchronous cells, prior to HU treatment, shows a significant overlap between MCM and Shu binding sites, supporting the interaction in unstressed conditions. We have addressed this discrepancy and its implications for our findings in the Discussion section.

It would be important to try this experiment without HU. It may be possible to observe Csm2 and Mcm moving together out of the ARS after release from alpha factor arrest.

If the two complexes do travel together, how do you explain the enrichment of Csm2 at ARS in an exponentially growing population of cells observed by ChIP seq (Fig.1A)?

We thank the reviewer regarding the observed enrichment of Csm2 at ARS during the exponential growth of cells as shown in ChIP-seq data from Figure 1A. In an exponentially growing population, cells are distributed across various stages of the cell cycle, with a significant proportion likely being in the S-phase where replication origins (ARS) are most active. This mixed population could result in varied binding interactions visible in our ChIP-seq analysis.

The ChIP-seq technique, while extremely useful for detecting protein-DNA interactions, is not inherently quantitative across the entire genome or all potential binding sites (ARSs). It is sensitive to strong interactions where the protein-DNA binding is more stable and thus more likely to be captured during the immunoprecipitation process. The PCR amplification step in ChIP-seq can disproportionately amplify these interactions, making it challenging to quantify how many cells in the population exhibit the interaction at any given time.

ok

ChIP seq seems to be performed with DNA fragments of up to 1 kb, which seems large to provide accurate results. Could you comment on this?

For each ChIP-seq sample, we ensured that the DNA fragments were around 200 bp. This was verified both before and after library preparation by gel electrophoresis and Tape Station analysis. This smaller fragment size is ideal for accurately identifying the locations of protein- DNA interactions. Additionally, it is important to note that the peaks identified in ChIP-seq data often represent a cluster of nearby binding sites rather than a single point of interaction. This clustering can give the impression of larger fragments but is a collection of multiple close interaction sites that are captured together due to the nature of the DNA-protein binding and the resolution of the sequencing technology (See Figure).

The fact that the DNA fragments were about 200 bp should be noted.

Figure legends could be simplified, e.g. by explaining the meaning of p-values only once.

We appreciate the Reviewer's astute observation. We have now simplified the Figure Legends.

ok

Introduction and Discussion: It is misleading to say that SHU has a role in repairing replicative damage, it is involved in DNA damage avoidance.

We have changed the wording from "repairing replicative damage" to "bypass of replicative damage" due to its function during replication.

ok

There are still some punctuation and other errors that need to be corrected (for example, (B-E) is repeated in the legend of Figure 1).

Also note that it was recently reported that HU arrest is more related to ROS production than RNR inhibition (Shaw et al, PNAS 2024, <https://www.pnas.org/lookup/suppl/doi:10.1073/pnas.2404470121>)

In conclusion, I think that the findings of an enrichment of SHU at ARS and of an interaction between SHU and initiation factors are very interesting. However, I found that there are not enough data to support that SHU travels with these factors during S phase and that several important experiments to support this hypothesis have not been attempted.

REFEREE CROSS-COMMENTS:

Referee 1:

(...)

My opinion is that there would be two solutions:

- Either modify the language to clearly state that with the current experiments they cannot determine if the SHU complex is moving with the replication machinery or not. I don't think this lowers the impact of the findings or the interest of the readership, and this is still an interesting future direction.
- If the authors want to proceed with the model that SHU is travelling with the replication machinery, then they should do the experiments suggested by reviewer 3.

Referee 2:

While I don't fully disagree with Reviewer 3's suggestion that doing more on SHU binding to ORCs and its progression with replication forks would strengthen the story, the current data in the manuscript already supports the authors' claims. Importantly, the observation that Shu-ORC/MCM binding mutants display lesion repair defects strongly suggests that SHU's binding to ORC/MCM may guide it to lesions encountered by the replisome.

(...)

In sum- without disagreeing with the reviewer, I feel that instead, a more in-depth interpretation and discussion of the existing observations will significantly sharpen the manuscript, highlighting the exciting questions it raises for future work.

Rebuttal letter

Fagunloye *et al.* (2024)

We would like to express our sincere gratitude for taking the time to thoroughly review our manuscript. We deeply value your thoughtful and constructive feedback, which has been instrumental in enhancing the quality and clarity of our work.

In response to your comments, we have carefully revised key sections of the text to address your concerns and incorporate your valuable suggestions.

Response to Reviewer Comments:

Referee #3:

1. It is better now, thanks, but looking at the violin plot in Fig.3, I still have the impression that CMG and GINS do not look that different, while the p-values do. This is not well explained. Maybe you should add numbers to the text for comparison?

- Thank you for the comment, we have now added, as suggested, numbers to the figure legend for clarity.

“The yeast Shu complex has evolutionary rates that correlate with CMG ($n=44$, $p < 0.001$), MCM ($n=24$, $p = 0.008$), and ORC ($n=24$, $p < 0.001$) complexes at levels higher than expected by chance when compared to a null distribution of 1000 permutations. Sample sizes for non-significant comparisons were GINS $n=16$, MTC $n=12$, and RAD51 $n=12$. Violin plots show higher evolutionary rate correlations between the yeast Shu complex and the replication initiation complexes when compared to the genome-wide background correlation, which is expected to be zero. Each dot represents co-evolution between two proteins compared to 1000 permuted nulls, the higher the correlation coefficient, the more significant the co-evolution. Significance is assessed by genome-wide permutation test contrasting observed Shu correlations against random protein sets”- Lines -1051-1060.

2. I don't understand this because an alpha factor arrest of Csm2-6HA cells released in G1 and followed in G2 is shown in Fig.1F to study the variation of the amount of Csm2-6HA. You can also use Psy3-6HA or Shu2-6HA. If there is a problem with synchrony, why don't you just do the experiment in alpha-factor arrested cells and nocodazole arrested cells (or with a mutation like *cdc28-as1*) as you did for your co-IP experiments (Fig.3G H I)? I think this experiment is very important to defend your traveling model. What if it turns out that SHU is enriched at ARS in G2?

- We appreciate the reviewer's insight into further substantiating our traveling model by addressing whether Shu complex components like Csm2 are enriched at ARS sites in G2. We have now softened our language of the Shu complex,

traveling with the fork in this paper. In the future, we will use the suggested experiments to address the questions.

3. Unfortunately, I don't think there is much evidence that SHU travels with the replication forks. In fact, I think that the fact that Csm2 is more highly expressed in S/G2 suggests that there might be "new" binding of free SHU to Mcm or Orc that is not associated with Mcm and Orc from the start of replication. In addition, the fact that the Csm2 Ip signal decreases at ARS after HU treatment, while forks are probably very slow (actually I have not found the concentration used in this experiment), does not fit with a permanent association of SHU with Mcm. It is also strange in the hypothesis that SHU and Mcm travel together that Mcm ChIP at ARS does not decrease after HU treatment (Fig. EV3D). ChIP or ChIP-Seq experiments in the different phase of the cell cycle and co-localization with Mcm/Orc is required.

- We are grateful for the reviewer's suggestions. Again, we have softened our language on the traveling model throughout the paper and proposed the above as our future direction.

“Although future experiments are needed to definitively demonstrate that the Shu complex travels with the replisome. Alternatively, the Shu complex may facilitate early S phase initiation at the origin to enable lesion bypass” -Lines 462-464.

“In the future, it will be important to determine if the Shu complex influences replication origin activation or replisome assembly and to delineate how the Shu complex moves or travels with the replication machinery to enable DNA lesion bypass” -Lines 488-491.

4. Again, a direct link between Csm2 traveling with Mcm after MMS treatment is lacking. It is still not shown why Csm2 is less recruited to ARS after HU and MMS treatment. In fact, Mcm is not less recruited after HU treatment (Fig. EV3D).

- We agree with the reviewer that additional experiments are necessary to clarify the relationship between Csm2 and Mcm during replication stress. We will prioritize these experiments in our future studies.

5. I don't understand this. It was possible to make the csm2-KRRRR strain and do ChIP experiments. There are 5 Psy3 mutations that could be used to do this experiment in a Psy3, shu2 or Csm2 tagged strain. The Psy3 mutations don't seem to have a negative effect on growth in the Y2H system or in the CAN1 and direct repeat strains. This is an important experiment and should be pursued.

- We thank the reviewer for this constructive suggestion. We agree that these experiments are both feasible and important for advancing the study. They will be prioritized in our future work to clarify the role of Csm2 and Psy3 mutations in Shu complex dynamics.

6. Since only Orc6 seems to show a Y2H interaction with Rad55 in Fig3D, it would be important to confirm that it is not mediated by SHU.

We have now performed the Y2H experiment to examine whether Rad55 and Orc6 interaction depends on the Shu complex members, Csm2 or Psy3. We find that loss of PSY3, but not CSM2, results in a reduced interaction between Rad55 and Orc6.

Ok, but I wonder if this new paragraph would be better placed in the first description of Fig. 3D.

- We thank the reviewer for the suggestion. We have carefully discussed this result in the results section of Figure 3D (as suggested) and in Figure 5C.

“Further analysis in (Fig. 5C) below, indicated that the loss of PSY3, but not CSM2, may lead to a modest reduction in the interaction between Rad55 and Orc6”-Lines 293-294.

“Since we find in Fig. 3D that Orc6 exhibits a Y2H interaction with Rad55, we examined whether Rad55 and Orc6 interaction depends on the Shu complex members, Csm2 or Psy3. Our result suggests that the loss of PSY3, but not CSM2, may modestly reduce the interaction between Rad55 and Orc6 (Fig. 5C), suggesting that Psy3 plays a subtle but important role in facilitating the Rad55-Orc6 interaction”-Lines 361-365.

Prof. Kara A Bernstein
University of Pennsylvania School of Medicine
Biochemistry and Biophysics
421 Curie Boulevard
BRB II/III Room 411
Philadelphia, PA 19104-6160

8th Jan 2025

Re: EMBOJ-2024-117270R1
The Shu complex interacts with replicative helicase to prevent mutations and aberrant recombination

Dear Kara and Katrin,

Thank you for submitting your final revised manuscript for our consideration. I am pleased to inform you that we have now accepted it for publication in The EMBO Journal.

With kind regards,

Hartmut
